# Learning from Noisy Preferences: A Semi-Supervised Learning Approach to Direct Preference Optimization

**Xinxin Liu, Ming Li, Zonglin Lyu, Yuzhang Shang, Chen Chen**
University of Central Florida

## Abstract

Human visual preferences are inherently multi-dimensional, encompassing aesthetics, detail fidelity, and semantic alignment. However, existing datasets provide only single, holistic annotations, resulting in severe label noise—images that excel in some dimensions but are deficient in others are simply marked as winner or loser. We theoretically demonstrate that compressing multi-dimensional preferences into binary labels generates conflicting gradient signals that misguide Diffusion Direct Preference Optimization (DPO). To address this, we propose Semi-DPO, a semi-supervised approach that treats consistent pairs as clean labeled data and conflicting ones as noisy unlabeled data. Our method starts by training on a consensus-filtered clean subset, then uses this model as an implicit classifier to generate pseudo-labels for the noisy set for iterative refinement. Experimental results demonstrate that Semi-DPO achieves state-of-the-art performance and significantly improves alignment with complex human preferences, without requiring additional human annotation or explicit reward models during training. We will release our code and models at: *https://github.com/L-CodingSpace/semi-dpo*.

## 1 Introduction

Diffusion models (Ho et al., 2020) have achieved remarkable success in text-to-image (T2I) generation (Ramesh et al., 2022; Pernias et al., 2023; Ramesh et al., 2021). However, aligning T2I diffusion models with human preferences typically requires training with separate reward models, creating significant computational bottlenecks (Wang et al., 2024; Wu et al., 2023a; Xu et al., 2023). To address this limitation, Diffusion-DPO (Wallace et al., 2024) adapts the direct preference optimization (DPO) approach from large language models (LLMs) (Rafailov et al., 2023b), eliminating the need for explicit reward models by optimizing directly on preference pairs. However, Diffusion-DPO overlooks a fundamental distinction: while human visual preferences are inherently multi-dimensional (Zhang et al., 2024), annotated datasets collapse this into binary choices. This destabilizes training by creating contradictory signals that penalize learning desirable attributes from "loser" images while simultaneously rewarding undesirable ones in "winner" images.

As shown in Figure 1: for the prompt "*A vast green grassland with blue sky and two white clouds, a mother cow and her calf both eating grass, natural landscape, extremely detailed, 4k resolution, perfect lighting, fine textures*", Image A may excel in semantic alignment and composition but appear aesthetically flat, while Image B may excel in texture but lack semantic alignment. When a human annotator is forced to choose, their judgment may hinge on a single dimension, yet the label is recorded as an overall preference. *This produces a noisy, contradictory signal, as the model is implicitly taught to prefer all attributes of the winning image, including its flaws.*

Collapsing multi-dimensional preferences into a single binary label introduces a critical challenge for Diffusion-DPO training: dimensional conflicts across the dataset cause conflicting gradient signals leading to suboptimal convergence. We formalize this phenomenon in our theoretical analysis 3.2. To address this challenge, we reformulate the problem through the lens of **learning with noisy labels (LNL)** (Zhang et al., 2016; Song et al., 2022). A dominant paradigm for tackling LNL is to reframe it as a **semi-supervised learning (SSL) problem**, where noisy samples are treated as an unlabeled set that requires relabeling. Within this paradigm, iterative self-training stands out as a

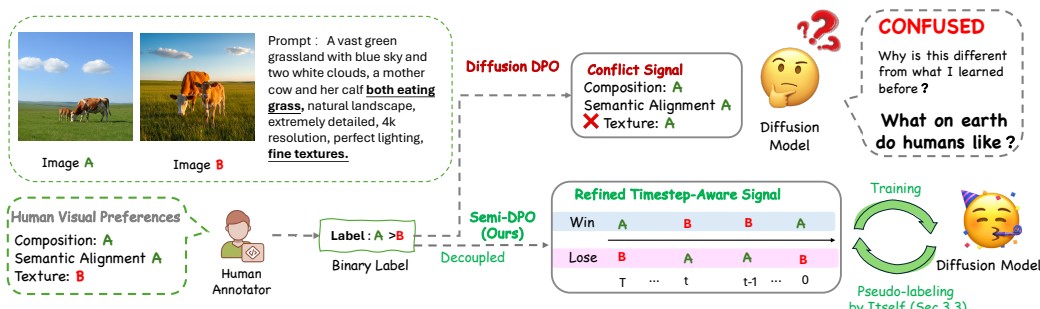

Figure 1: **Resolving label noise from multi-dimensional preferences.** Standard Diffusion-DPO learns from a noisy, conflict signal created when a binary label (A ≻ B) collapses multi-dimensional preferences (e.g., A's composition vs. B's texture). Our method, Semi-DPO, resolves this by decoupling the conflict into refined timestep aware signals via self-training, leading to a more robust alignment.

powerful and widely adopted methodology. It involves training a classifier on a small clean set and using it to generate pseudo-labels for the larger unlabeled set, progressively refining the model (Xie et al., 2020; Li et al., 2020; Arazo et al., 2019). Inspired by this, we treat dimensionally conflicted samples as our unlabeled set. The central question for this approach then becomes *which model should serve as the classifier for generating these pseudo-labels*.

Our answer lies within the *diffusion model* itself. We leverage the fact that the DPO loss compels the model to distinguish between preferred and dispreferred samples, effectively turning it into an implicit preference classifier. This inherent capability allows the diffusion model to generate fine-grained pseudo-labels for dimensionally conflicted data without requiring any architectural changes. Furthermore, echoing research (Hertz et al., 2022) on the hierarchical nature of the diffusion process, where early stages govern global composition and later stages refine local details. This hierarchy allows us to reframe a single, conflicting preference label (e.g., A's good composition vs. B's good texture) as a series of non-conflicting, timestep-conditional preferences. Therefore, we apply this implicit classifier across the diffusion timeline. This transforms a single noisy preference signal into fine-grained, timestep-conditioned pseudo-labels, effectively decoupling noisy conflicting signals.

To this end, we propose **Semi-DPO**, a two-stage framework. In the first stage, **Multi-Reward Consensus**, we partition the dataset by using a consensus of diverse, pre-trained reward models to filter the data. A preference pair is added to a clean labeled set only if all models unanimously agree with the human label; otherwise, it is assigned to a noisy, unlabeled set requiring relabeling. This process designates about 21% of the Pick-a-Pic V2 dataset as clean. In the second stage, **Iterative self-training**, we initially train the model on the clean labeled dataset. The trained model then generates timestep-conditional pseudo-labels for the noisy unlabeled dataset. To mitigate confirmation bias and model drift (Zhang et al., 2021; Cascante-Bonilla et al., 2021; Xie et al., 2020), only high-confidence pseudo-labels are selected for retraining with a composite objective anchored to the clean set. This creates a virtuous cycle, synergistically improving the model's alignment with multi-dimensional human preferences.

In summary, our main contributions are: (1) We provide a theoretical analysis proving that conflicting dimensional signals in holistic labels cause conflicting gradient signal during Diffusion-DPO training, leading to suboptimal convergence. (2) We propose Semi-DPO, a novel self-training framework that reframes the preference alignments as semi-supervised learning under label noise, leveraging timestep-conditional pseudo-labeling to generate fine-grained signals that decouple conflicting preference dimensions. (3) We demonstrate through extensive experiments that Semi-DPO achieves state-of-the-art performance, significantly improving the model's ability to generate images that align with complex, multi-dimensional human preferences without extra annotation costs or fine-tuning with an explicit reward model.

## 2 RELATED WORKS

**Diffusion Models and Diffusion Alignment.** In recent years, diffusion models (Ho et al., 2020; Song & Ermon, 2019; Song et al., 2020) have achieved remarkable success in text-to-image gen-

eration. These models are traditionally trained on large-scale text-image datasets scraped from the web. However, they are not well-aligned with human preferences. To achieve better alignment, T2I adapted Reinforcement Learning from Human Feedback (RLHF) (Ouyang et al., 2022) from the LLM domain (Dai et al., 2023; Miao et al., 2024; Clark et al., 2024), which requires training an explicit reward model on human preference data to guide diffusion model optimization Li et al. (2024a); Xu et al. (2023). However, developing reliable reward models remains computationally expensive and requires large-scale annotated datasets (Xu et al., 2023; Wu et al., 2023a; Wang et al., 2024), creating a significant bottleneck. Inspired by Direct Preference Optimization (DPO) (Rafailov et al., 2023b) in LLMs, recent work has adapted this approach to T2I alignment, eliminating the need for an explicit reward model. Methods like Diffusion-DPO (Wallace et al., 2024) directly optimize diffusion models on human-annotated preference pairs by maximizing the relative probability of preferred images. Follow-up work on Diffusion-DPO falls into two categories: offline (Li et al., 2024b; Zhu et al., 2025; Lee et al., 2025; Hong et al., 2024; Li et al., 2026) and online methods (Liang et al., 2025; Black et al., 2023; Zhang et al., 2025; Yang et al., 2024). *We provide detailed comparisons between online and offline diffusion DPO methods in Appendix 6.5 and discuss how Semi-DPO relates to existing work in Appendix 6.6.*

**Noise Data & Semi-Supervised Learning.** The success of deep learning models largely depends on large-scale, high-quality annotated datasets. However, common data collection methods—such as web scraping and crowdsourcing (e.g., Amazon Mechanical Turk)—inevitably introduce label errors (Song et al., 2022). Due to their high capacity, deep neural networks can memorize these incorrect labels, which degrades generalization (Zhang et al., 2016; Song et al., 2022). To address this challenge, a dominant paradigm reframes learning with noisy labels (LNL) as a semi-supervised learning (SSL) problem (Arazo et al., 2019). This approach partitions training data into a clean labeled dataset and a noisy unlabeled dataset (Li et al., 2020; Han et al., 2018; Yu et al., 2019; Wei et al., 2020). This partitioning strategy has been applied in several influential frameworks. Coteaching (Han et al., 2018) trains two networks that select small-loss samples for each other. Noisy Student Training (Xie et al., 2020) employs self-training where a teacher generates pseudo-labels for a larger, noised student model, enabling it to learn more robust representations.

## 3 METHOD

### 3.1 PRELIMINARIES

**Diffusion Models.** Diffusion Models are latent variable models designed to learn the reverse of a fixed, $T$-step Markovian noising process. The forward process, $q$, is defined as $q(\mathbf{x}_t \mid \mathbf{x}_{t-1}) := \mathcal{N}(\mathbf{x}_t; \sqrt{1 - \beta_t}\mathbf{x}_{t-1}, \beta_t\mathbf{I})$, which admits a closed-form sampling distribution at any timestep $t$: $q(\mathbf{x}_t \mid \mathbf{x}_0) = \mathcal{N}(\mathbf{x}_t; \sqrt{\bar{\alpha}_t}\mathbf{x}_0, (1 - \bar{\alpha}_t)\mathbf{I})$, where $\alpha_t = 1 - \beta_t$ and $\bar{\alpha}_t = \prod_{i=1}^{t} \alpha_i$.

The model learns the reverse process, $p_\theta(\mathbf{x}_{t-1} \mid \mathbf{x}_t, \boldsymbol{c})$, by training a network $\epsilon_\theta(\mathbf{x}_t, t, \boldsymbol{c})$ to predict the noise component $\epsilon$ from a noised sample $\mathbf{x}_t$. This is achieved by optimizing a simplified objective on the negative log-likelihood:

$$\mathcal{L}_{\mathrm{DM}} = \mathbb{E}_{t,\mathbf{x}_0,\epsilon} \left[ w(t) \left\| \epsilon - \epsilon_\theta \left( \sqrt{\bar{\alpha}_t}\mathbf{x}_0 + \sqrt{1 - \bar{\alpha}_t}\epsilon, t, \boldsymbol{c} \right) \right\|_2^2 \right] \tag{1}$$

Generation is performed via ancestral sampling, starting from $\mathbf{x}_T \sim \mathcal{N}(0, \mathbf{I})$ and iteratively applying the learned reverse transition.

**Reinforcement Learning from Human Feedback (RLHF).** A prevalent approach for model alignment is Reinforcement Learning from Human Feedback (RLHF). For text-to-image (T2I) models, human preference data is collected as paired comparisons $(\mathbf{x}_0^w, \mathbf{x}_0^l, \boldsymbol{c})$, where $\mathbf{x}_0^w$ and $\mathbf{x}_0^l$ represent the preferred ("winning") and dispreferred ("losing") final images for a given text prompt $\boldsymbol{c}$.

First, a reward function $r(\mathbf{x}_0, \boldsymbol{c})$ is trained to model these preferences using the Bradley-Terry model, where the likelihood of preferring $\mathbf{x}_0^w$ over $\mathbf{x}_0^l$ is given by:

$$p_{\mathrm{BT}}\big(\mathbf{x}_0^w \succ \mathbf{x}_0^l \mid \boldsymbol{c}\big) = \sigma\big(r(\mathbf{x}_0^w, \boldsymbol{c}) - r(\mathbf{x}_0^l, \boldsymbol{c})\big) \tag{2}$$

where $\sigma(\cdot)$ is the sigmoid function. Subsequently, the diffusion model policy $p_\theta$ is optimized to maximize the expected reward, regularized by a KL-divergence term to prevent large deviations from a reference policy $p_{\mathrm{ref}}$, with $\beta > 0$ controls the KL penalty strength:

$$\mathcal{L}_{\mathrm{RLHF}} = \mathbb{E}_{p_\theta(\mathbf{x}_0|\boldsymbol{c})} \left[ r(\mathbf{x}_0, \boldsymbol{c}) \right] - \beta \, \mathbb{D}_{\mathrm{KL}} \left[ p_\theta(\mathbf{x}_{0:T} \mid \boldsymbol{c}) \, \| \, p_{\mathrm{ref}}(\mathbf{x}_{0:T} \mid \boldsymbol{c}) \right] \tag{3}$$

**Direct Preference Optimization for Diffusion Models.** Direct Preference Optimization (DPO) simplifies RLHF by re-parameterizing the reward function directly in terms of the policy and reference models, thus bypassing the need for an explicit reward model. When adapting this concept to diffusion models by framing the denoising process as a Markov Decision Process, the implicit, time-step-wise reward function can be expressed as:

$$r(\mathbf{x}_{t-1}, \boldsymbol{c}) = \beta \log \frac{p_\theta(\mathbf{x}_{t-1} \mid \mathbf{x}_t, \boldsymbol{c})}{p_{\text{ref}}(\mathbf{x}_{t-1} \mid \mathbf{x}_t, \boldsymbol{c})} \tag{4}$$

Substituting this reward definition into the Bradley-Terry likelihood yields the conceptual Diffusion-DPO loss, which is optimized directly with respect to the policy parameters $\theta$:

$$\mathcal{L}_{\text{DPO-Diffusion}} = -\mathbb{E}\left[\log \sigma\left(\beta \log \frac{p_\theta(\mathbf{x}_{t-1}^w \mid \mathbf{x}_t^w, \boldsymbol{c})}{p_{\text{ref}}(\mathbf{x}_{t-1}^w \mid \mathbf{x}_t^w, \boldsymbol{c})} - \beta \log \frac{p_\theta(\mathbf{x}_{t-1}^l \mid \mathbf{x}_t^l, \boldsymbol{c})}{p_{\text{ref}}(\mathbf{x}_{t-1}^l \mid \mathbf{x}_t^l, \boldsymbol{c})}\right)\right] \tag{5}$$

The expectation is taken over the preference dataset $\mathcal{D}$ and timesteps $t \in [1, T]$, where the noisy states $\mathbf{x}_t$ are sampled from the forward noising process.

## 3.2 Conflicting Gradients Signal from Multi-dimensional Preferences

**Diffusion-DPO Gradient Formulation.** To understand the training dynamics, we begin by decomposing the per-sample, per-timestep DPO gradient, $\nabla_\theta \mathcal{L}_{\text{DPO}}^{(t)}$.

$$\nabla_\theta \mathcal{L}_{\text{DPO}}^{(t)} = - \underbrace{(1 - \sigma(z_\theta^{(t)})) \cdot \beta}_{f_\theta^{(t)}} \underbrace{\left(\nabla_\theta \log p_\theta(\mathbf{x}_{t-1}^w | \mathbf{x}_t^w, c) - \nabla_\theta \log p_\theta(\mathbf{x}_{t-1}^l | \mathbf{x}_t^l, c)\right)}_{\Delta\phi_\theta^{(t)}} \tag{6}$$

where $z_\theta^{(t)} := \beta\left(\log \frac{p_\theta(\mathbf{x}_{t-1}^w | \mathbf{x}_t^w, c)}{p_{\text{ref}}(\mathbf{x}_{t-1}^w | \mathbf{x}_t^w, c)} - \log \frac{p_\theta(\mathbf{x}_{t-1}^l | \mathbf{x}_t^l, c)}{p_{\text{ref}}(\mathbf{x}_{t-1}^l | \mathbf{x}_t^l, c)}\right)$. This decomposition shows that the Diffusion-DPO update adjusts the model's parameters along the direction of the feature difference, $\Delta\phi_\theta^{(t)}$, aiming to increase the log-probability of the preferred sample relative to the dispreferred one. While this update mechanism is effective for clean preference pairs, its behavior becomes problematic in the presence of multi-dimensional conflicts. Detailed derivation is provided in Appendix 6.1. We now analyze how these conflicts impact the Diffusion-DPO training process.

**The Source of Conflicting Signals in Preference Optimization.** In this section, we provide a theoretical analysis for the training instability outlined previously. We demonstrate how collapsing multi-dimensional preferences into binary labels mathematically guarantees the presence of conflicting gradient signals, leading to suboptimal convergence. Our analysis begins by partitioning the dataset based on whether a sample's preference along a specific dimension aligns with its holistic label. For a given dimension $k$ (e.g., composition), let the reward difference be $\Delta r_k := r_k(\mathbf{x}_0^w, c) - r_k(\mathbf{x}_0^l, c)$. This partitions the dataset into an alignment set $\mathcal{A}_k$ (where the dimensional preference matches the holistic label, $\Delta r_k > 0$) and a conflict set $\mathcal{C}_k$ (where the dimensional preference opposes the holistic label, $\Delta r_k < 0$), occurring with probabilities $p_{a,k}$ and $p_{c,k}$, respectively.

To analyze the training dynamics, we define the per-sample gradient $g_\theta^{(t)} := -f_\theta^{(t)} \cdot \Delta\phi_\theta^{(t)}$, and the oracle direction, $v_k(\theta, t) := \text{sign}(\Delta r_k)\Delta\phi_\theta^{(t)}$. The oracle direction represents the ideal update direction for improving dimension $k$. For an aligned pair in $\mathcal{A}_k$, $v_k$ is the standard Diffusion-DPO update direction. However, for a conflicting pair in $\mathcal{C}_k$, $v_k$ points in the opposite direction, representing the corrective update that should have been applied for this specific dimension.

We then examine the inner product. The inner product defined as $\left\langle -g_\theta^{(t)}, v_k(\theta, t) \right\rangle$. This quantity measures how well the actual gradient update aligns with the ideal oracle direction. A positive value indicates that the update is making progress on dimension $k$, while a negative value indicates the update is actively degrading performance on that dimension. *Therefore, the variance of this inner product serves as a direct mathematical measure for the severity of conflicting signals that cause training instability.*

As we prove in Appendix 6.2, this variance is bounded below by:

$$\text{Var}[\langle -g_\theta^{(t)}, v_k(\theta, t)\rangle] \geq p_{a,k} p_{c,k} \cdot (m_{a,k}^{(t)} + m_{c,k}^{(t)})^2 \tag{7}$$

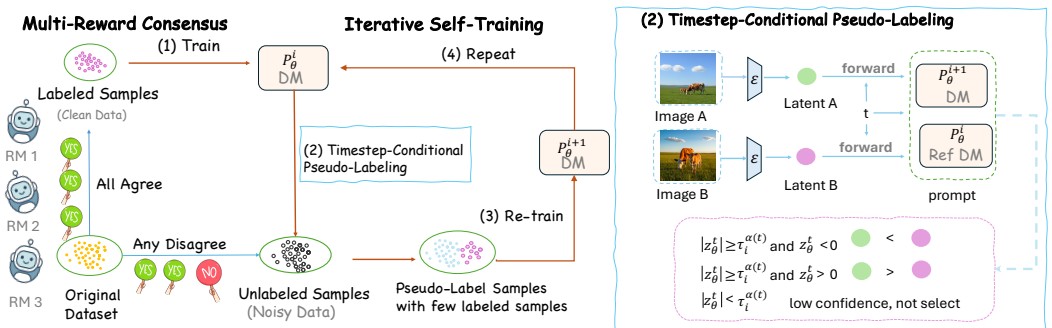

Figure 2: Semi-DPO framework for resolving label noise. Stage 1 (**Multi-Reward Consensus**): A committee of reward models partitions the original dataset into a small, clean labeled dataset (based on unanimous agreement) and a large, noisy unlabeled dataset . Stage 2 (**Iterative Self-Training**): (1) An initial model is trained on the clean labeled set. (2) This model then generates pseudo-labels for the noisy unlabeled set. A pseudo-label is accepted if its confidence score (the logit magnitude $|z_\theta^t|$) exceeds a dynamic threshold $\tau_i^{\alpha(t)}$. The sign of the logit, $\text{sign}(z_\theta^t)$, determines how the new label is applied: a positive sign keeps the original "winner" and "loser" assignment for the image pair, while a negative sign swaps them. (3-4) The model is then retrained on the high-confidence pseudo-labels and the original clean set, and this cycle is repeated until convergence.

where $m_{a,k}^{(t)}$ and $m_{c,k}^{(t)}$ are the expected magnitudes of the gradient updates conditioned on the alignment and conflict sets, respectively. This inequality reveals the mechanism behind the instability. The product term $p_{a,k}p_{c,k}$ is greater than zero if and only if a conflict set exists ( $p_{c,k} > 0$ ), which mathematically guarantees that the training signal must contain updates that are both aligned with and opposed to the oracle direction. This co-existence gives rise to gradients with inconsistent directions updates from $\mathcal{C}_k$ that directly oppose the "oracle" updates desired for dimension $k$.

Conflicting gradient signals force the model's parameters to oscillate, with the update direction frequently reversing. This behavior introduces two critical issues: (1) it renders the learning process highly inefficient, as progress from one step is largely negated by the next, impeding the minimization of the loss function. (2) this constant directional conflict makes the training instability, and the model struggles to find a consistent optimization path, leading to suboptimal convergence.

### 3.3 THE SEMI-DPO FRAMEWORK

Our theoretical analysis shows that collapsing multi-dimensional conflicts into single binary labels generates conflicting gradient signals, causing suboptimal convergence. To address this, we reframe the alignment task as a semi-supervised learning (SSL) problem designed to handle these noisy labels. We detail our proposed framework in the following section.

**Data Partitioning via Multi-Reward Consensus.** To obtain a clean labeled set ($\mathcal{D}_{\text{labeled}}$) for stable cold-start training, we filter the original dataset $\mathcal{D}$ using a multi-reward consensus approach. This approach is motivated by existing work that reveals strong correlations between widely-used reward models and different dimensions of human preference Zhang et al. (2024) (see Table 6 and Figure 4). For instance, CLIP Score (Radford et al., 2021) demonstrates a strong correlation with the semantic alignment dimension, while Aesthetic Score (Schuhmann et al., 2022a) is highly correlated with the aesthetics dimension (detailed explanation provided in Appendix 6.8). We employ a set of $K$ pre-trained models $\{r_k\}_{k=1}^K$. A preference pair $(c, x_0^w, x_0^l)$ is included in $\mathcal{D}_{\text{labeled}}$ only if all reward models unanimously agree with the holistic label, i.e., $\forall k, \Delta r_k = r_k(x_0^w, c) - r_k(x_0^l, c) > 0$. The remaining data, which contains dimensional conflicts, forms the noisy unlabeled set $\mathcal{D}_{\text{unlabeled}}$. This filtering step yields a high-quality dataset that provides an unambiguous initial gradient direction.

**Timestep-Conditional Pseudo-Labeling.** Our pseudo-labeling strategy stems from a key insight: the DPO loss function is equivalent to a binary cross-entropy loss. This means the training process implicitly trains a binary classifier at each timestep to distinguish between the denoising predictions associated with the preferred and dispreferred samples. Consequently, the per-timestep margin, $z_\theta^{(t)}$, functions as the logit for this implicit classifier. This provides a principled signal for self-training, where the sign of the logit, $\text{sign}(z_\theta^{(t)})$, determines the predicted preference, and its magnitude, $|z_\theta^{(t)}|$, serves as the confidence score (see Figure 2, right).

**Dynamic Timestep-Conditional Thresholding.** Our experiments find that the model's confidence and prediction accuracy are not uniform over the diffusion timesteps (Table 7). Therefore, instead of a single fixed threshold for pseudo-labeling, we employ a dynamic strategy. We partition the timeline into $N$ distinct intervals, $\{I_j\}_{j=1}^N$, and assign a unique threshold to each. This dynamic threshold, $\tau_{i-1}^{\alpha(t)}$, is updated at each training iteration and is specific to the interval $\alpha(t)$ containing timestep $t$. Consequently, a pseudo-label is used for retraining only if its confidence score $|z_{\theta_{i-1}}^{(t)}|$ exceeds the corresponding threshold for its specific time interval, as shown in Figure 2 right.

**Iterative Self-Training with a Composite Objective.** The iterative self-training process begins with a "cold-start" phase. We first train an initial model, $p_\theta^0$, using only the clean labeled set $\mathcal{D}_{\text{labeled}}$. This step is crucial as it provides a stable and reliable foundation for the self-training loop. The training objective at this stage consists solely of the anchor loss (defined in Eq. 9). For each subsequent iteration $i$, we leverage the model from the previous step, $p_\theta^{i-1}$, to generate pseudo-labels for the noisy unlabeled set $\mathcal{D}_{\text{unlabeled}}$. The new model, $p_\theta^i$, is then trained using the composite objective (defined in Eq. 8) that combines a stable, anchoring signal from the clean set with a learning signal from high-confidence pseudo-labels (see Figure 2, left). These principles are formalized in our composite objective for the iterative refinement stage ($i > 0$):

$$\mathcal{L}_{\text{Semi-DPO}}^{(i)}(\theta) = \mathcal{L}_{\text{labeled}}(\theta) + \mathcal{L}_{\text{unlabeled}}^{(i)}(\theta) \tag{8}$$

The two loss components are defined as:

**Anchor Loss ($\mathcal{L}_{\text{labeled}}$).** The standard Diffusion-DPO loss on the clean labeled set, which acts as a ground-truth regularizer to prevent model drift.

$$\mathcal{L}_{\text{labeled}}(\theta) = \mathbb{E}_{(c,x_0^w,x_0^l)\sim\mathcal{D}_{\text{labeled}}}\left[-\log\sigma(z_\theta^{(t)})\right] \tag{9}$$

**Pseudo-Label Loss ($\mathcal{L}_{\text{unlabeled}}^{(i)}$).** The DPO loss on a filtered subset of the noisy data, using pseudo-labels generated by the model from the previous iteration ($p_\theta^{i-1}$).

$$\mathcal{L}_{\text{unlabeled}}^{(i)}(\theta) = \mathbb{E}_{(c,x_0^w,x_0^l)\sim\mathcal{D}_{\text{unlabeled}}}\left[\mathbb{I}(|z_{\theta_{i-1}}^{(t)}| > \tau_{i-1}^{\alpha(t)})\cdot\left(-\log\sigma(\hat{z}_\theta^{(t)})\right)\right] \tag{10}$$

In Eq. 10, the indicator function $\mathbb{I}(\cdot)$ filters for high-confidence predictions using the dynamic threshold $\tau_{i-1}^{\alpha(t)}$. The new preference pair $(x_{\text{pseudo}}^w, x_{\text{pseudo}}^l)$ used to compute the loss term $\hat{z}_\theta^{(t)}$ is determined by the sign of the previous model's logit, $z_{\theta_{i-1}}^{(t)}$. A positive sign retains the original label, while a negative sign swaps the "winner" and "loser" images.

This pseudo-labeling mechanism is the core of our solution to the inflated gradient variance problem identified in Section 3.2. The variance originates from dimensional conflicts within $\mathcal{D}_{\text{unlabeled}}$ where a sample's holistic label provides a supervisory signal that opposes the ideal gradient for a specific attribute. Our method resolves these conflicts at a granular, timestep-conditional level. By re-labeling pairs based on the model's own learned preference-the sign of the logit $z_{\theta_{i-1}}^{(t)}$-it actively corrects the noisy original annotations. This process effectively reduces the proportion of conflicting samples (the term $p_{c,k}$ in our analysis) that generate gradients with inconsistent directions. Consequently, this self-correction mitigates the source of gradient conflict, leading to a more consistent and effective training signal from the noisy dataset.

## 4 EXPERIMENTS

### 4.1 EXPERIMENTAL SETTING

**Datasets and Models**. We select SD1.5 (Rombach et al., 2022) and SDXL (Podell et al., 2023) as our base model. We train Semi-DPO on the Pick-a-Pic V2 (Kirstain et al., 2023) dataset. After excluding approximately 12% of ties pairs, there are 851,293 preference pairs across 58,960 unique prompts in the training dataset.

**Baselines**. We compare Semi-DPO against strong baselines, including the original base models (SD1.5 Rombach et al. (2022), SDXL Podell et al. (2023)) and state-of-the-art alignment methods

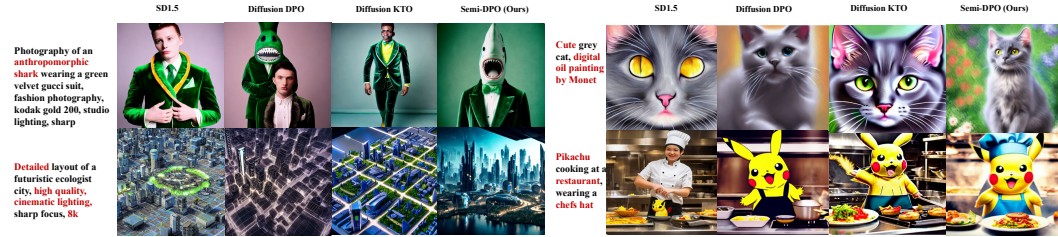

Figure 3: Sample images generated by different models for various prompts.

such as Diffusion-DPO Rafailov et al. (2023a), Diffusion-KTO Li et al. (2024b), MaPO Hong et al. (2024), and InPO Lu et al. (2025). All comparisons are conducted using their officially released checkpoints.

**Training Details**. During the multi-reward consensus stage, we employ five proxy reward models to filter the dataset: PickScore (Kirstain et al., 2023), HPS v2 (Wu et al., 2023a), CLIP Score (Radford et al., 2021), the LAION Aesthetics Classifier (Schuhmann et al., 2022a), and ImageReward (Xu et al., 2023). This filtering process, which is further detailed in Appendix 6.8 and Section 3.3, yields a clean, consensus-labeled dataset of 176,999 pairs, with the remainder classified as noisy. We then split this clean dataset into a training portion of 173,007 pairs and a test portion of 3,992 pairs, which is used to evaluate model accuracy after each training iteration (See Appendix 6.9). An ablation study investigating the impact of using different numbers of reward models is presented in Section 4.3. Additional training details can be found in Appendix 6.9.

**Evaluation.** Following the protocol from (Li et al., 2024b), we evaluate our Semi-DPO method on the SD1.5 model. The model's performance is assessed using a suite of established metrics: ImageReward, PickScore, HPS v2, the LAION Aesthetics Classifier, CLIP Score, and GenEval (Ghosh et al., 2023). Furthermore, to demonstrate Semi-DPO's ability to generate outputs that align with multi-dimensional human preferences, we employ the Multi-dimensional Preference Score (MPS) (Zhang et al., 2024).

## 4.2 Alignment Result

**Qualitative Result.** Figure 3 presents a qualitative comparison between Semi-DPO and baseline models using the same prompt, demonstrating our method's significant improvements in text-alignment, detail fidelity, and aesthetics. For instance, given the prompt, *"a photo of Pikachu cooking at a restaurant, wearing a chef's hat,"* Semi-DPO is the only method that successfully generates Pikachu with the specified chef's hat, highlighting its superior text-alignment. We provide more visualization comparison results in Figure 6.

**Quantitative Result.** Tables 1 demonstrate that Semi-DPO consistently outperforms baselines on both SD1.5 and SDXL, excelling in multi-reward metrics and win-loss rates (refer to Tables 8. Consistent with prior studies (Zhang et al., 2025; Lee et al., 2025), we further evaluate on specialized benchmarks: GenEval (Ghosh et al., 2023) for object-focused generation and T2I-CompBench (Huang et al., 2025) for compositional generation. Results in Table 2 and 3 confirm Semi-DPO's significant advantage, validating its capability in generating complex, high-fidelity images.

## 4.3 Ablation Study

**Iterations.** To validate the contribution of our iterative self-training process, we conducted an ablation study on the key stages of Semi-DPO, as shown in Table 4. Our hypothesis is that each iteration enables the model to generate better pseudo-labels, leading to further performance gains. We evaluated the performance of different iterations: Iter0, the initial model trained only on the high-quality clean dataset; Iter1, the model after the first round of pseudo-labeling and retraining; and Iter2, the model after the second round of pseudo-labeling and retraining.

Our ablation study results substantiate our hypothesis. We find that each iteration yields a model capable of generating more reliable pseudo-labels, which in turn enhances the performance of the subsequent model. Specifically, we observe significant performance improvements from Iter0 to Iter1 and from Iter1 to Iter2. However, our results indicate these performance gains diminish and stabilize after the second iteration. We therefore conclude two rounds of self-training are sufficient for convergence, striking an effective balance between performance and computational efficiency.(see Appendix 6.9 for a detailed cost analysis).

Table 1: Reward Score comparisons on HPS v2, Parti-Prompt and Pick-a-Pic V2 datasets. We compare baselines across both SD1.5 and SDXL architectures. Best results are in **boldface**.

| Model | Dataset | Method | ImageReward | HPSv2.1 | PickScore | Aesthetic | CLIP | MPS |
|---|---|---|---|---|---|---|---|---|
| **SD1.5** | HPS v2 | SD1.5 | 0.139 | 0.246 | 20.862 | 5.578 | 0.293 | 12.211 |
| | | Diff-DPO | 0.339 (+0.200) | 0.259 (+5.3%) | 21.308 (+2.1%) | 5.714 (+2.4%) | 0.297 (+1.4%) | 12.739 (+4.3%) |
| | | Diff-KTO | 0.690 (+0.551) | 0.284 (+15.4%) | 21.454 (+2.8%) | 5.803 (+4.0%) | 0.298 (+1.7%) | 13.016 (+6.6%) |
| | | Semi-DPO | **0.816** (+0.677) | **0.287** (+16.7%) | **21.945** (+5.2%) | **5.899** (+5.8%) | **0.299** (+2.0%) | **13.514** (+10.7%) |
| | Parti Prompt | SD1.5 | 0.194 | 0.254 | 21.284 | 5.358 | 0.270 | 9.754 |
| | | Diff-DPO | 0.352 (+0.158) | 0.262 (+3.1%) | 21.520 (+1.1%) | 5.443 (+1.6%) | 0.272 (+0.7%) | 10.135 (+3.9%) |
| | | Diff-KTO | 0.615 (+0.421) | 0.279 (+9.8%) | 21.594 (+1.5%) | 5.552 (+3.6%) | **0.277** (+2.6%) | 10.202 (+4.6%) |
| | | Semi-DPO | **0.798** (+0.604) | **0.284** (+11.8%) | **21.964** (+3.2%) | **5.706** (+6.5%) | 0.276 (+2.2%) | **10.771** (+10.4%) |
| | Pick-a-Pic V2 | SD1.5 | 0.085 | 0.250 | 20.566 | 5.421 | 0.273 | 9.635 |
| | | Diff-DPO | 0.297 (+0.212) | 0.261 (+4.4%) | 20.948 (+1.9%) | 5.549 (+2.4%) | 0.279 (+2.2%) | 10.144 (+5.3%) |
| | | Diff-KTO | 0.629 (+0.544) | 0.281 (+12.4%) | 21.064 (+2.4%) | 5.659 (+4.4%) | **0.281** (+2.9%) | 10.226 (+6.1%) |
| | | Semi-DPO | **0.801** (+0.716) | **0.288** (+15.2%) | **21.524** (+4.7%) | **5.801** (+7.0%) | **0.281** (+2.9%) | **11.030** (+14.5%) |
| **SDXL** | HPS v2 | SDXL | 0.839 | 0.280 | 22.611 | 6.127 | 0.301 | 14.472 |
| | | Diff-DPO | 1.070 (+0.231) | 0.298 (+6.4%) | **22.953** (+1.5%) | 6.163 (+0.6%) | **0.307** (+2.0%) | 14.831 (+2.5%) |
| | | MaPO | 0.960 (+0.121) | 0.292 (+4.3%) | 22.682 (+0.3%) | 6.244 (+1.9%) | 0.303 (+0.7%) | 14.635 (+1.1%) |
| | | Semi-DPO | **1.095** (+0.256) | **0.301** (+7.5%) | 22.816 (+0.9%) | **6.269** (+2.3%) | 0.302 (+0.3%) | **15.001** (+3.7%) |
| | Parti Prompt | SDXL | 0.736 | 0.272 | 22.407 | 5.766 | 0.278 | 11.255 |
| | | Diff-DPO | **1.046** (+0.310) | 0.288 (+5.9%) | 22.681 (+1.2%) | 5.813 (+0.8%) | **0.286** (+2.9%) | **11.792** (+4.8%) |
| | | MaPO | 0.855 (+0.119) | 0.280 (+2.9%) | 22.412 (+0.0%) | **5.939** (+3.0%) | 0.280 (+0.7%) | 11.431 (+1.6%) |
| | | Semi-DPO | 0.983 (+0.247) | **0.291** (+7.0%) | **22.715** (+1.4%) | 5.845 (+1.4%) | 0.281 (+1.1%) | 11.633 (+3.4%) |
| | Pick-a-Pic V2 | SDXL | 0.716 | 0.282 | 22.031 | 6.055 | 0.283 | 11.809 |
| | | Diff-DPO | 0.953 (+0.237) | 0.299 (+6.0%) | **22.467** (+2.0%) | 6.031 (-0.4%) | **0.292** (+3.2%) | 12.210 (+3.4%) |
| | | MaPO | 0.852 (+0.136) | 0.290 (+2.8%) | 22.091 (+0.3%) | 6.189 (+2.2%) | 0.286 (+1.1%) | 11.887 (+0.7%) |
| | | Semi-DPO | **1.056** (+0.340) | **0.304** (+7.8%) | 22.352 (+1.5%) | **6.210** (+2.6%) | 0.287 (+1.4%) | **12.548** (+6.3%) |

Table 2: Quantitative results on GenEval (Ghosh et al., 2023) with 50 inference steps. Comparison between SD1.5 and SDXL baselines. Best results are in **boldface**.

| Base | Method | Single | Two | Counting | Colors | Position | Color_attr | Overall |
|---|---|---|---|---|---|---|---|---|
| SD1.5 | SD1.5 | 95.62 | 37.63 | 37.81 | 74.73 | 3.50 | 4.75 | 42.34 |
| | Diff-DPO | 96.88 | 39.90 | 38.75 | 75.53 | 3.30 | 3.75 | 43.00 |
| | Diff-KTO | 97.50 | 35.35 | 36.25 | 79.79 | **7.00** | 6.00 | 43.65 |
| | InPO | 95.00 | 45.45 | **45.00** | **82.98** | 4.00 | 8.00 | 46.74 |
| | Semi-DPO | **98.75** | **49.75** | 42.19 | 77.93 | 6.00 | **9.25** | **47.31** |
| SDXL | SDXL | 98.12 | 75.25 | 43.75 | **89.63** | 11.25 | 15.75 | 55.63 |
| | Diff-DPO | **99.38** | **82.58** | 49.06 | 85.11 | 13.05 | 18.55 | 58.02 |
| | MaPO | 96.56 | 66.41 | 40.00 | 84.31 | 10.75 | 18.75 | 52.80 |
| | InPO | 97.50 | 74.75 | 46.25 | 84.04 | 10.00 | 18.00 | 55.09 |
| | Semi-DPO | 97.50 | 80.81 | **50.00** | 86.17 | **14.00** | **22.00** | **58.41** |

Table 3: Quantitative Results on T2I-CompBench++ Huang et al. (2025). Comparison between SD1.5 and SDXL baselines. Best results are in **boldface**.

| Base | Method | Color | Shape | Texture | 2D-Spa. | 3D-Spa. | Numer. | Non-Spa. | Complex |
|---|---|---|---|---|---|---|---|---|---|
| SD1.5 | Original | 0.378 | 0.362 | 0.417 | 0.123 | 0.297 | 0.449 | 0.310 | 0.300 |
| | Diffusion-DPO | 0.409 | 0.366 | 0.425 | 0.134 | 0.312 | 0.454 | 0.312 | 0.304 |
| | Diffusion-KTO | 0.465 | 0.416 | 0.466 | 0.157 | **0.341** | 0.461 | **0.314** | 0.309 |
| | InPO | **0.482** | 0.424 | **0.493** | 0.159 | **0.341** | 0.468 | **0.314** | 0.319 |
| | Semi-DPO | 0.471 | **0.433** | **0.493** | **0.183** | 0.340 | **0.481** | 0.310 | **0.320** |
| SDXL | Original | 0.5833 | 0.4782 | 0.5211 | 0.1936 | 0.3319 | 0.4874 | 0.3137 | 0.3327 |
| | Diff-DPO | **0.6941** | **0.5311** | **0.6127** | **0.2153** | 0.3686 | 0.5304 | **0.3178** | 0.3525 |
| | MaPO | 0.6090 | 0.5043 | 0.5485 | 0.1964 | 0.3473 | 0.5015 | 0.3154 | 0.3229 |
| | InPO | 0.6409 | 0.5067 | 0.5720 | 0.2107 | 0.3671 | 0.5221 | 0.3129 | 0.3674 |
| | Semi-DPO | 0.6624 | 0.5079 | 0.5727 | 0.2133 | **0.3723** | **0.5410** | 0.3154 | **0.3728** |

Table 4: Ablation study of the iterative self-training process on SD1.5.

| Dataset | Method | ImageReward | HPS v2.1 | PickScore | Aesthetic | CLIP | MPS |
|---------|--------|-------------|----------|-----------|-----------|------|-----|
| **HPS v2** | Semi-DPO (Iter0) | 0.569 | 0.269 | 21.493 | 5.806 | **0.300** | 13.039 |
| | Semi-DPO (Iter1) | 0.798 | 0.284 | 21.892 | **5.902** | **0.300** | 13.495 |
| | Semi-DPO (Iter2) | **0.816** | **0.287** | **21.945** | 5.899 | 0.299 | **13.514** |
| **Parti Prompt** | Semi-DPO (Iter0) | 0.557 | 0.272 | 21.679 | 5.548 | 0.275 | 10.386 |
| | Semi-DPO (Iter1) | 0.779 | 0.283 | 21.929 | 5.691 | **0.277** | 10.743 |
| | Semi-DPO (Iter2) | **0.798** | **0.284** | **21.964** | **5.706** | 0.276 | **10.771** |
| **Pick-a-Pic V2** | Semi-DPO (Iter0) | 0.563 | 0.273 | 21.153 | 5.660 | **0.282** | 10.554 |
| | Semi-DPO (Iter1) | 0.789 | 0.287 | 21.490 | 5.794 | **0.282** | 11.018 |
| | Semi-DPO (Iter2) | **0.801** | **0.288** | **21.524** | **5.801** | 0.281 | **11.030** |

**Number of Reward Models for Consensus Filtering.** To determine the optimal number of proxy reward models for multi-reward consensus, we ablated consensus sizes of two to five models on SD1.5. As shown in Table 5, performance consistently improves across all metrics as the committee size increases. Notably, incorporating a greater number of reward models reduces the bias from individual evaluators and not only boosts performance on the metrics used for filtering but also enhances the model's generalization ability to other preference evaluators not included in the consensus (MPS).We therefore adopt a five-model consensus to ensure the initial model is trained on the most reliable clean data.

Table 5: **Ablation study on the number of reward models for consensus filtering.** This study evaluates the performance of a model trained on data filtered by a consensus of an increasing number of reward models (from two to five) to determine the optimal configuration. Metrics included in the consensus committee are marked in green, while metrics that are evaluated but not used for filtering are marked in red. The baseline SD1.5 and its results are marked in gray. Best results for each dataset are in **bold**.The results demonstrate that performance consistently improves across all metrics and datasets as the consensus committee grows.

| Dataset | Method | CLIP | Aesthetic | HPSv2 | ImageReward | PickScore | MPS |
|---------|--------|------|-----------|-------|-------------|-----------|-----|
| **HPS v2** | SD1.5 | 0.293 | 5.578 | 0.246 | 0.139 | 20.860 | 12.211 |
| | CLIP+Aes | 0.299 | 5.759 | 0.263 | 0.442 | 21.438 | 12.891 |
| | CLIP+Aes+HPS | 0.299 | 5.775 | 0.265 | 0.459 | 21.430 | 12.927 |
| | CLIP+Aes+HPS+IR | 0.299 | 5.789 | 0.268 | 0.524 | 21.464 | 13.018 |
| | CLIP+Aes+HPS+IR+Pick | **0.300** | **5.806** | **0.269** | **0.569** | **21.493** | **13.039** |
| **Parti Prompt** | SD1.5 | 0.270 | 5.358 | 0.254 | 0.194 | 21.284 | 9.754 |
| | CLIP+Aes | 0.273 | 5.501 | 0.265 | 0.444 | 21.620 | 10.280 |
| | CLIP+Aes+HPS | 0.273 | 5.519 | 0.268 | 0.470 | 21.639 | 10.331 |
| | CLIP+Aes+HPS+IR | 0.274 | 5.530 | 0.269 | 0.513 | 21.651 | 10.363 |
| | CLIP+Aes+HPS+IR+Pick | **0.275** | **5.548** | **0.272** | **0.557** | **21.679** | **10.386** |
| **Pick-a-Pic V2** | SD1.5 | 0.273 | 5.421 | 0.250 | 0.085 | 20.566 | 9.635 |
| | CLIP+Aes | 0.279 | 5.609 | 0.267 | 0.417 | 21.099 | 10.399 |
| | CLIP+Aes+HPS | 0.280 | 5.627 | 0.270 | 0.469 | 21.143 | 10.454 |
| | CLIP+Aes+HPS+IR | 0.281 | 5.632 | 0.271 | 0.499 | 21.127 | 10.477 |
| | CLIP+Aes+HPS+IR+Pick | **0.282** | **5.660** | **0.273** | **0.563** | **21.153** | **10.554** |

## 5 CONCLUSION

In this work, we theoretically demonstrate that collapsing multi-dimensional preferences into binary labels generates conflicting gradients, hindering Diffusion-DPO optimization. To resolve this, we propose Semi-DPO, a semi-supervised framework that identifies clean data via Multi-Reward Consensus and corrects noisy labels through iterative self-training. By leveraging timestep-conditional pseudo-labels, our method effectively decouples conflicting dimensions. Experiments confirm that Semi-DPO achieves state-of-the-art performance, significantly improving alignment with complex human preferences without requiring additional annotations or explicit reward models. Moreover, our findings highlight the intrinsic capability of diffusion models to serve as their own latent reward models, paving the way for more scalable, self-contained alignment frameworks that are robust to real-world data imperfections.

ETHICS STATEMENT

Our research enhances text-to-image model alignment while relying solely on publicly available resources, specifically the Pick-a-Pic V2 (Kirstain et al., 2023) dataset, SD1.5 Rombach et al. (2022) and SDXL (Podell et al., 2023). In line with responsible research practices, we acknowledge the broader societal implications and potential for misuse associated with generative technologies.

REPRODUCIBILITY STATEMENT

To ensure reproducibility, we will release our code. Our method is detailed in Section 3.2, and our training hyperparameters are documented in Appendix 6.9 to allow for the replication of our results.

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

## 6 APPENDIX

### 6.1 DERIVATION OF THE DIFFUSION-DPO GRADIENT

We derive the gradient of the per-timestep Diffusion-DPO loss, $\mathcal{L}_{\text{DPO}}^{(t)}(\theta)$, with respect to the model parameters $\theta$. The loss is defined as:

$$\mathcal{L}_{\text{DPO}}^{(t)}(\theta) := -\log \sigma \left( \beta \left[ \log \frac{p_\theta(\mathbf{x}_{t-1}^w | \mathbf{x}_t^w, c)}{p_{\text{ref}}(\mathbf{x}_{t-1}^w | \mathbf{x}_t^w, c)} - \log \frac{p_\theta(\mathbf{x}_{t-1}^l | \mathbf{x}_t^l, c)}{p_{\text{ref}}(\mathbf{x}_{t-1}^l | \mathbf{x}_t^l, c)} \right] \right) \tag{11}$$

To simplify the notation, let $h_\theta^{(t)} = \beta \left[ \log \frac{p_\theta(\mathbf{x}_{t-1}^w | \mathbf{x}_t^w, c)}{p_{\text{ref}}(\mathbf{x}_{t-1}^w | \mathbf{x}_t^w, c)} - \log \frac{p_\theta(\mathbf{x}_{t-1}^l | \mathbf{x}_t^l, c)}{p_{\text{ref}}(\mathbf{x}_{t-1}^l | \mathbf{x}_t^l, c)} \right]$. Applying the chain rule and using the identity $\frac{d}{dx}(-\log \sigma(x)) = -(1 - \sigma(x))$, we have:

$$\begin{aligned}
\nabla_\theta \mathcal{L}_{\text{DPO}}^{(t)}(\theta) &= \nabla_\theta \left( -\log \sigma(h_\theta^{(t)}) \right) \\
&= \frac{d}{dh_\theta^{(t)}} \left( -\log \sigma(h_\theta^{(t)}) \right) \nabla_\theta h_\theta^{(t)} \\
&= -\left( 1 - \sigma(h_\theta^{(t)}) \right) \nabla_\theta h_\theta^{(t)} \\
&= -(1 - \sigma(h_\theta^{(t)})) \nabla_\theta \left[ \beta \left( \log \frac{p_\theta(\mathbf{x}_{t-1}^w | \mathbf{x}_t^w, c)}{p_{\text{ref}}(\mathbf{x}_{t-1}^w | \mathbf{x}_t^w, c)} - \log \frac{p_\theta(\mathbf{x}_{t-1}^l | \mathbf{x}_t^l, c)}{p_{\text{ref}}(\mathbf{x}_{t-1}^l | \mathbf{x}_t^l, c)} \right) \right] \\
&= -(1 - \sigma(h_\theta^{(t)})) \cdot \beta \left( \nabla_\theta \log p_\theta(\mathbf{x}_{t-1}^w | \mathbf{x}_t^w, c) - \nabla_\theta \log p_\theta(\mathbf{x}_{t-1}^l | \mathbf{x}_t^l, c) \right)
\end{aligned} \tag{12}$$

### 6.2 PROOF OF VARIANCE INFLATION

For clarity, we first restate the relevant definitions from Section 3.2. Let $\mathcal{D}$ be the dataset of preference tuples $d = (c, x_0^w, x_0^l)$. For any sample $d \in \mathcal{D}$ and a given preference dimension $k$, we define the reward difference as $\Delta r_k(d) := r_k(x_0^w, c) - r_k(x_0^l, c)$. This induces a partition of the dataset into an alignment set $\mathcal{A}_k := \{d \in \mathcal{D} \mid \Delta r_k(d) > 0\}$ and a conflict set $\mathcal{C}_k := \{d \in \mathcal{D} \mid \Delta r_k(d) < 0\}$, with respective probabilities $p_{a,k} := P(d \in \mathcal{A}_k)$ and $p_{c,k} := P(d \in \mathcal{C}_k)$. For a given sample $d$ and timestep $t$, we define the per-sample gradient $g_\theta^{(t)} := -f_\theta^{(t)} \cdot \Delta\phi_\theta^{(t)}$, the oracle direction $v_k(\theta, t) := \text{sign}(\Delta r_k) \cdot \Delta\phi_\theta^{(t)}$, and the inner product $\xi_t := \left\langle -g_\theta^{(t)}, v_k(\theta, t) \right\rangle$. Finally, we define the conditional expected magnitudes as

$$m_{a,k}^{(t)} := \mathbb{E}\left[ f_\theta^{(t)} \cdot \left\| \Delta\phi_\theta^{(t)} \right\|_2^2 \mid \mathcal{A}_k \right] \text{ and } m_{c,k}^{(t)} := \mathbb{E}\left[ f_\theta^{(t)} \cdot \left\| \Delta\phi_\theta^{(t)} \right\|_2^2 \mid \mathcal{C}_k \right]$$

By the law of total variance, we can decompose the variance of $\xi_t$ based on whether a sample is in the alignment set $\mathcal{A}_k$ or the conflict set $\mathcal{C}_k$:

$$\text{Var}[\xi_t] = \underbrace{\mathbb{E}[\text{Var}[\xi_t \mid Z]]}_{\text{Intra-group variance}} + \underbrace{\text{Var}(\mathbb{E}[\xi_t \mid Z])}_{\text{Inter-group variance}} \tag{13}$$

where $Z$ is a random variable indicating membership in $\{\mathcal{A}_k, \mathcal{C}_k\}$.

Since variance is non-negative, the total variance is bounded below by the inter-group variance term:

$$\text{Var}[\xi_t] \geq \text{Var}(\mathbb{E}[\xi_t \mid Z]) \tag{14}$$

We now compute the $\text{Var}(\mathbb{E}[\xi_t \mid Z])$. The conditional expectations of $\xi_t$ are:

- $\mathbb{E}[\xi_t \mid \mathcal{A}_k] = \mathbb{E}[f_\theta^{(t)} \cdot (+1) \cdot \|\Delta\phi_\theta^{(t)}\|_2^2 \mid \mathcal{A}_k] = m_{a,k}^{(t)}$

- $\mathbb{E}[\xi_t \mid \mathcal{C}_k] = \mathbb{E}[f_\theta^{(t)} \cdot (-1) \cdot \|\Delta\phi_\theta^{(t)}\|_2^2 \mid \mathcal{C}_k] = -m_{c,k}^{(t)}$

The $\mathrm{Var}\left(\mathbb{E}\left[\xi_t \mid Z\right]\right)$ can now be calculated using its definition, $\mathrm{Var}(Y) = \mathbb{E}[Y^2] - (\mathbb{E}[Y])^2$. Let $Y = \mathbb{E}[\xi_t \mid Z]$. Then:

$$
\begin{aligned}
\mathrm{Var}(\mathbb{E}[\xi_t \mid Z]) &= p_{a,k}(m_{a,k}^{(t)})^2 + p_{c,k}(-m_{c,k}^{(t)})^2 - (p_{a,k}m_{a,k}^{(t)} - p_{c,k}m_{c,k}^{(t)})^2 \\
&= p_{a,k}(1 - p_{a,k})(m_{a,k}^{(t)})^2 + p_{c,k}(1 - p_{c,k})(m_{c,k}^{(t)})^2 + 2p_{a,k}p_{c,k}m_{a,k}^{(t)}m_{c,k}^{(t)} \\
&= p_{a,k}p_{c,k}\left((m_{a,k}^{(t)})^2 + (m_{c,k}^{(t)})^2 + 2m_{a,k}^{(t)}m_{c,k}^{(t)}\right) \\
&= p_{a,k}p_{c,k}(m_{a,k}^{(t)} + m_{c,k}^{(t)})^2
\end{aligned}
\tag{15}
$$

where we used the fact that $p_{a,k} + p_{c,k} = 1$.

Thus, the total variance has a lower bound determined by the conflict:

$$
\mathrm{Var}[\xi_t] \geq p_{a,k} \cdot p_{c,k} \cdot (m_{a,k}^{(t)} + m_{c,k}^{(t)})^2
\tag{16}
$$

This proves that any non-zero conflict mass ($p_{c,k} > 0$) introduces a variance term that grows quadratically with the sum of the conflicting and aligned update magnitudes.

## 6.3 THE USE OF LARGE LANGUAGE MODELS

Large Language Models (LLMs) such as GPT were used solely for language polishing and clarity improvements in the writing of this paper. All technical content, dataset design, experimental results, and analyses were created by the authors. The models were not used to generate ideas, data, or experimental outcomes.

## 6.4 LIMITATIONS

Like many classical SSL methods (Xie et al., 2020; Zhang et al., 2021; Cascante-Bonilla et al., 2021), our approach requires multiple cycles of pseudo-labeling and model retraining. However, as detailed in Appendix 6.9, this iterative process does not result in higher computational costs; in fact, our method is more efficient than the single-stage baseline (132 vs. 192 GPU hours) due to the effective filtering of noisy data. Therefore, the primary limitation lies not in computational resources, but in the operational complexity of managing a multi-stage pipeline. Future research could explore unified, single-pass frameworks to mitigate this engineering overhead.

## 6.5 COMPARISON OF ONLINE AND OFFLINE DPO PARADIGMS IN IMAGE GENERATION

The methodologies for Direct Preference Optimization (DPO) in diffusion models can be broadly categorized into offline and online approaches. Offline sampling-based DPO in T2I, the paradigm under which Semi-DPO operates, **utilizes a static, pre-collected dataset** of human preferences for the entirety of the training process. This approach is computationally efficient and offers greater stability and reproducibility, as it fine-tunes the model in a single stage on a fixed dataset. However, its primary limitation is that the model's ultimate performance is fundamentally constrained by the diversity and quality of the initial preference data; it cannot learn beyond the scope of the examples it is given (Wallace et al., 2024; Zhu et al., 2025; Lee et al., 2025; Hong et al., 2024).

In contrast, online sampling-based DPO in T2I involves dynamically generating new preference data during the training loop. In this setup, the diffusion model iteratively produces new images that are then evaluated, typically by an auxiliary reward model, to **create new preference pairs for continued training**. While this allows the model to learn continuously and potentially surpass the quality of the initial dataset, it introduces significant computational overhead and complexity. Furthermore, online methods risk overfitting to the biases of the reward model and depend on having a reliable reward signal available throughout the resource-intensive training process (Liang et al., 2025; Black et al., 2023; Zhang et al., 2025; Yang et al., 2024).

## 6.6 COMPARISON WITH EXISTING METHODS

**Offline DPO.** Unlike existing offline approaches that focus on modifying algorithms (Zhu et al., 2025; Li et al., 2024b; Hong et al., 2024) or generating new datasets through multiple reward models

for re-annotation and training (Lee et al., 2025), Semi-DPO addresses the noise label problem caused by multi-dimensional preference conflicts in DPO datasets. We reclassify the dataset into clean labeled data and noisy unlabeled data. Following the principle of semi-supervised learning, Semi-DPO leverages both labeled and unlabeled data to achieve superior performance compared to using labeled data alone. This approach maximizes the utilization of existing datasets.

Our theoretical analysis and empirical results serve a dual purpose: validating our central claim that collapsing multi-dimensional preferences into single binary labels introduces a significant noisy label problem, and demonstrating that a diffusion model trained with the DPO loss can correct these noisy labels by acting as its own implicit classifier.

**Latent Reward Model.** Semi-DPO leverages a key property of the Diffusion-DPO framework: its loss function transforms the diffusion model into an implicit latent reward model. The principle is straightforward: the DPO objective compels the model to increase the relative probability of preferred samples over dispreferred ones. To achieve this, the model must learn to distinguish between "better" and "worse" latent representations at every timestep, a capability that serves as an inherent reward signal.

This approach contrasts with existing latent reward model methods (Zhang et al., 2025; Ding et al., 2025). Those methods typically require architectural modifications to construct an explicit reward model that is then trained separately. In contrast, the latent reward model in Semi-DPO is the original diffusion model itself. This implicit method is more efficient and requires no architectural changes.

## 6.7 FUTURE WORK: AN ONLINE EXTENSION OF THE SEMI-DPO PARADIGM

Pixel-space reward models (e.g., ImageReward) are constrained by gradient propagation issues, which restrict their effective training signal to only the final stages of the diffusion process. Latent reward models overcome this limitation by providing a deep training signal across all timesteps (from $t = 999$ down to $t = 1$ ), allowing them to guide the entire generation process.

However, existing latent reward models (Zhang et al., 2025; Ding et al., 2025) have a significant limitation: they are architecturally specific. They must share a latent space with the diffusion model they are training, which requires a shared VAE encoder. As different generative models (e.g., SD1.5 and SDXL) use different VAEs, a latent reward model trained for one is incompatible with another. This gives them great power but prevents the plug-and-play versatility of pixel-space models.

Our work offers a path to resolve this trade-off. At its core, a diffusion model trained with the DPO loss functions as an implicit latent reward model, learning to assign preference labels at each timestep. This insight allows for a powerful online extension of the Semi-DPO paradigm:

- **Cold-Start:** An initial model is trained on a small, multi-dimensionally consistent dataset to learn the basics of human preference.

- **Online Training:** During online training, the model inherits our iterative self-training philosophy. To begin the $(i+1)$-th iteration, new data is generated by the model from iteration $i$ and then labeled by an ensemble of implicit reward models composed of the models from iterations $i$ and $i - 1$.

By developing this iterative, self-training online method, we would no longer need to train a new, bespoke latent reward model for each T2I architecture. Instead, the model would correct itself through an internal process. This presents a path toward a universal, model-agnostic alignment strategy that captures the deep-signal benefits of latent-space rewards without being constrained by their architectural limitations.

## 6.8 MOTIVATION BY MULTI-REWARD SELECTION

In our methods (see Section 3.3), we employ a committee of five proxy reward models PickScore (Kirstain et al., 2023), HPS V2 (Wu et al., 2023a), CLIP (Radford et al., 2021), LAION Aesthetics Classifier (Schuhmann et al., 2022b), and ImageReward (Xu et al., 2023) for data filtering. This approach is motivated by the study that introduced the Multi-dimensional Preference Score (MPS) (Zhang et al., 2024), which constructed a dataset reflecting real human preferences by

Table 6: The evaluation of MPS and scoring functions for the prediction of multi-dimensional human preferences(%). **Copied from MPS (Zhang et al., 2024)**.

| ID | Preference Model | Overall | Aesthetics | Alignment | Detail |
|----|------------------|---------|-----------|-----------|--------|
| 1 | CLIP score Radford et al. (2021) | 63.67 | 68.14 | 82.69 | 61.71 |
| 2 | Aesthetic Score Schuhmann et al. (2022a) | 62.85 | 82.85 | 69.36 | 60.34 |
| 3 | ImageReward Xu et al. (2023) | 67.45 | 74.79 | 75.27 | 58.31 |
| 4 | HPS Wu et al. (2023b) | 65.51 | 73.86 | 73.86 | 62.05 |
| 5 | PickScore Kirstain et al. (2023) | 69.52 | 70.95 | 70.92 | 56.74 |
| 6 | HPS v2 Wu et al. (2023a) | 65.51 | 73.86 | 73.87 | 62.06 |
| 7 | MPS (Zhang et al., 2024) | **74.24** | **83.86** | **83.87** | **85.18** |

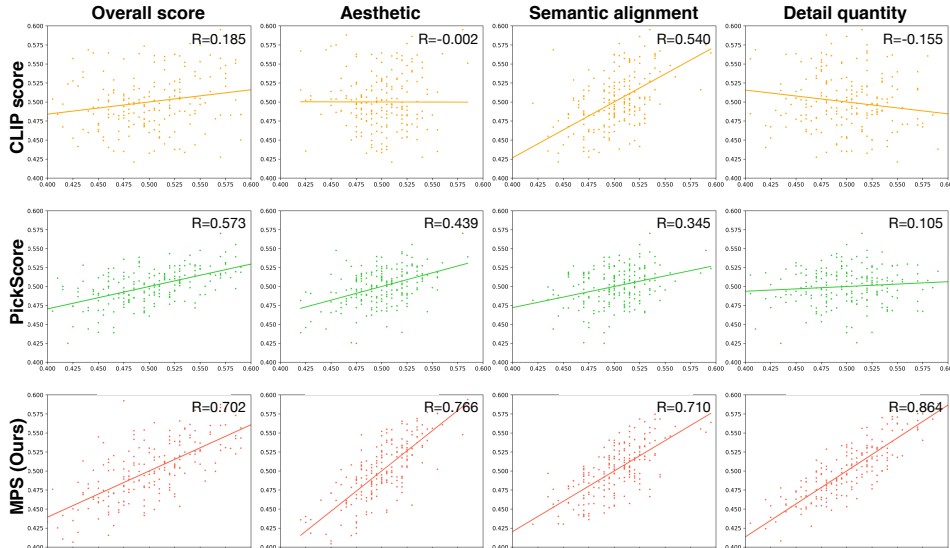

Figure 4: **Correlation between real user preferences and model predictions**. The x-axis of each subplot represents the annotated real human preferences, and the y-axis denotes the model's predictions. We examine three models: CLIP score, PickScore, and MPS. Each subplot is annotated with the calculated correlation coefficient R-value, where a higher R-value indicates a closer alignment of the model's predictions with actual human preferences. **Figure reproduced from (Zhang et al., 2024).**.

ensuring image source diversity and having human annotators perform pairwise comparison scoring across four dimensions: aesthetics, detail quality, semantic alignment, and overall assessment.

They compared the performance of multiple existing reward models against their proposed MPS on a multi-dimensional preference dataset, revealing a core phenomenon: existing reward models exhibit significant specialization when predicting human preferences (see Table 6 and Figure 4. For instance, CLIP Score (Radford et al., 2021) shows a strong correlation with the semantic alignment dimension, while Aesthetic Score (Schuhmann et al., 2022a) is highly correlated with the aesthetics dimension. Meanwhile, models such as HPSv2 (Wu et al., 2023a), ImageReward (Xu et al., 2023), and PickScore (Kirstain et al., 2023) show higher consistency with the overall score. This finding indicates that no single model can comprehensively evaluate image quality.

As shown in Table 6 and Figure 4 (**Both Table 6 and Figure 4 are reproduced from MPS paper. They are Tab. 4 and Fig. 5 in their original paper.**), the MPS model shows the strongest correlation with multi-dimensional human preferences. However, a publicly available version capable of providing scores for individual dimensions was unavailable at the time of our work. Therefore, to ensure our initial training data was reliable and free from label noise caused by dimensional conflicts, we used five distinct preference models in concert, selecting only the data points that all models agreed upon to form a high-quality, dimensionally consistent subset. Figure 5 provides a visual comparison of these two subsets, highlighting the prevalence of label noise and dimensional conflicts within the data rejected by our consensus mechanism.To validate the effectiveness of this five-model filtering strategy, we also conducted an ablation study to investigate the impact of using different numbers of reward models 4.3.

| Winner | Loser | Winner | Loser |
|--------|-------|--------|-------|

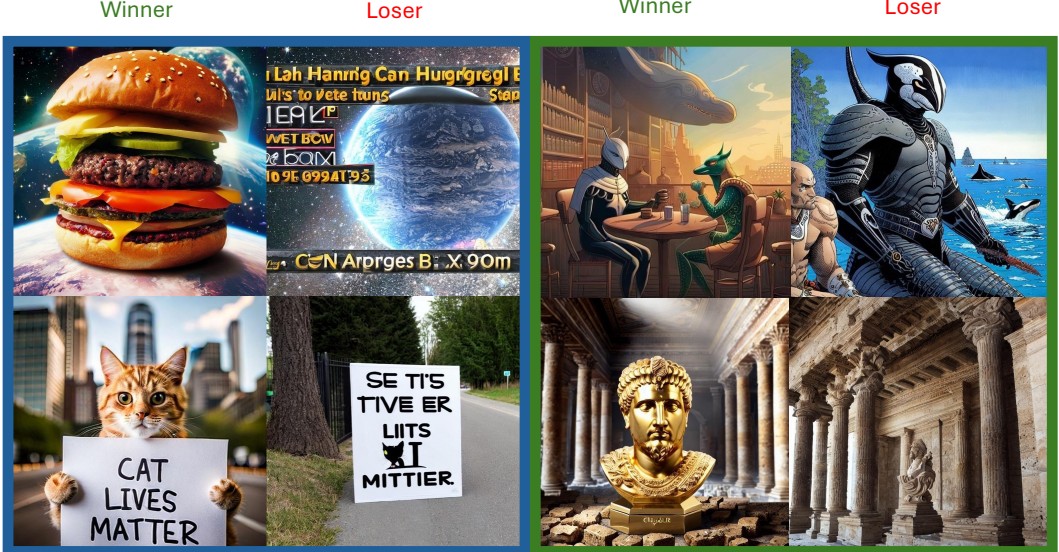

Figure 5: **Qualitative comparison of samples from the labeled dataset (Left, Blue) and the unlabeled dataset (Right, Green).** The annotations **Winner** (green) and **Loser** (red) indicate the human preference labels. The corresponding full prompts are: **Top-Left:** "Cosmic hamburger in space"; **Bottom-Left:** "Cat with a sign that says 'Cat lives matter' "; **Top-Right:** "An empowering view of a orca warrior wearing royal robe, sitting in a cafe drinking coffee next to a kangaroo warrior with an eye scar, menacing, by artist Philippe Druillet and Tsutomu Nihei, volumetric lighting, detailed shadows, extremely detailed"; **Bottom-Right:** "A wide angle photo of large gold head Caesar on display in a smokey roman villa burning, 18mm smoke filled room debris, gladiator, floor mosaics fire smoke, a photo, roman, a digital rendering, inside the roman colosseum, brick, indoor, plants overgrown outstanding detail, room flooded with water, in front of a building, by claude-joseph vernet, luxury hotel".

## 6.9 TRAINING DETAILS

For SD1.5, the training was utilizing a total of 32 NVIDIA A100 40GB GPUs for distributed training. We configured a local batch size of 4 for each GPU and performed gradient accumulation over 4 steps, which resulted in a global batch size of 512. Iteration 0 uses a learning rate of $4 \times 10^{-9}$ and is trained for 1,600 steps, while iterations 1 and 2 use a learning rate of $4 \times 10^{-10}$ and are trained for 4,000 steps each. The DPO parameter was set to $\beta = 2500$ for all iterations, in line with the hyperparameters specified in the official Diffusion-DPO (Wallace et al., 2024) code repository. All iterations incorporate a linear warmup over the first 400 steps.

To implement our dynamic thresholds for pseudo-labeling (as mentioned in Section 3.3, we first partition the diffusion timeline ($t \in [0, 999]$) into ten discrete intervals

Table 7: The model's prediction accuracy, evaluated on a clean test set, varies across the diffusion timeline.

| Timesteps | 50 | 150 | 250 | 350 | 450 | 550 | 650 | 750 | 850 | 950 |
|-----------|----|-----|-----|-----|-----|-----|-----|-----|-----|-----|
| **Accuracy (%)** | 72 | 73 | 73 | 72 | 72 | 71 | 69 | 67 | 65 | 59 |

(e.g., 0-100, 100-200). We initially set the threshold for each interval at the 80th percentile of its confidence scores to ensure a consistent number of samples are initially selected. However, when we evaluated this strategy on our accuracy test portion (3,992 pairs), we found that the model's prediction accuracy was not uniform across different timestep intervals (see Table 7; specifically, for timesteps greater than 650, its accuracy dropped below 70%. To mitigate confirmation bias from these less reliable labels, for any interval failing to meet the 70% accuracy level, we raised its confidence threshold. Although this adjustment reduces the number of pseudo-labeled samples from later stages of the diffusion process, it ensures the model is primarily trained on labels that are both high-confidence and high-accuracy.

**Computational Efficiency.** Under the setup described above, completing the first two stages (Iter 0 and Iter 1) requires approximately **132 GPU hours**. This is significantly more efficient than the

standard Diffusion-DPO baseline, which necessitates **192 GPU hours**. This efficiency gain stems directly from the dynamic thresholding mechanism described previously, which effectively filters out low-confidence samples from the noisy unlabeled dataset. While extending training to Iter 2 increases the total cost to 228 GPU hours, the performance gains are marginal compared to Iter 1 (see Table 4, indicating that the Iter 0 + 1 configuration offers the optimal balance between training efficiency and performance.

**Inference & Memory Costs.** Since our final model retains the exact same architecture as the base model, the inference time and memory footprint are identical to the baseline, incurring zero additional computational cost during deployment.

## 6.10 ADDITIONAL QUANTITATIVE RESULTS

Table 8: Win rate (%) comparisons on Pick-a-Pic V2, HPS V2 and Parti-Prompt datasets. Comparisons are performed for both SD1.5 and SDXL baselines.

| Base | Dataset | Method1 | Method2 | ImageReward | HPS v2 | PickScore | Aesthetic | CLIP | MPS |
|------|---------|---------|---------|-------------|--------|-----------|-----------|------|-----|
| **SD1.5** | HPS v2 | Diff-DPO | SD1.5 | 60.80% | 70.10% | 75.70% | 68.10% | 54.70% | 67.20% |
| | | Diff-KTO | SD1.5 | 77.80% | **90.40%** | 74.10% | 72.20% | 55.30% | 68.60% |
| | | Semi-DPO | SD1.5 | **79.80%** | 88.10% | **87.40%** | **78.80%** | **55.80%** | **73.70%** |
| | | Semi-DPO | Diff-DPO | **74.60%** | **83.00%** | **77.40%** | **68.40%** | **52.20%** | **66.40%** |
| | | Semi-DPO | Diff-KTO | **56.10%** | **52.80%** | **72.90%** | **60.60%** | **51.20%** | **61.70%** |
| | Parti Prompt | Diff-DPO | SD1.5 | 58.90% | 64.90% | 67.90% | 63.20% | 50.80% | 62.80% |
| | | Diff-KTO | SD1.5 | 69.80% | **83.40%** | 65.20% | 69.10% | **57.30%** | 59.60% |
| | | Semi-DPO | SD1.5 | **75.80%** | **83.40%** | **78.80%** | **80.00%** | 56.00% | **71.70%** |
| | | Semi-DPO | Diff-DPO | **71.90%** | **78.80%** | **70.80%** | **74.00%** | **55.00%** | **65.70%** |
| | | Semi-DPO | Diff-KTO | **60.20%** | **54.80%** | **70.40%** | **65.70%** | 48.60% | **66.20%** |
| | Pick-a-Pic V2 | Diff-DPO | SD1.5 | 63.20% | 69.90% | 74.60% | 66.00% | 58.00% | 66.00% |
| | | Diff-KTO | SD1.5 | 75.50% | 85.80% | 73.40% | 71.80% | **60.90%** | 62.70% |
| | | Semi-DPO | SD1.5 | **79.10%** | **87.30%** | **85.20%** | **80.70%** | 60.60% | **77.00%** |
| | | Semi-DPO | Diff-DPO | **72.90%** | **82.10%** | **76.30%** | **72.60%** | **53.50%** | **69.00%** |
| | | Semi-DPO | Diff-KTO | **60.20%** | **59.50%** | **73.00%** | **64.50%** | **50.40%** | **70.20%** |
| **SDXL** | HPS v2 | Diff-DPO | SDXL | **69.10%** | 80.90% | **69.90%** | 54.30% | **59.70%** | 61.90% |
| | | MaPO | SDXL | 64.60% | 78.90% | 55.80% | **67.10%** | 55.40% | 56.10% |
| | | Semi-DPO | SDXL | 68.90% | **82.90%** | 60.40% | 62.50% | 52.40% | **64.70%** |
| | | Semi-DPO | Diff-DPO | **52.90%** | **52.80%** | 41.90% | **61.20%** | 42.50% | **54.40%** |
| | | Semi-DPO | MaPO | **60.10%** | **65.50%** | **57.20%** | 49.40% | 46.60% | **60.90%** |
| | Parti Prompt | Diff-DPO | SDXL | **69.20%** | 76.80% | **63.80%** | 55.70% | **61.10%** | 63.60% |
| | | MaPO | SDXL | 61.30% | 70.20% | 48.30% | 70.80% | 55.10% | 57.10% |
| | | Semi-DPO | SDXL | 68.40% | **80.90%** | 58.90% | **71.30%** | 56.80% | **70.70%** |
| | | Semi-DPO | Diff-DPO | **51.40%** | **58.00%** | 45.10% | **70.20%** | 46.30% | **59.40%** |
| | | Semi-DPO | MaPO | **62.20%** | **71.00%** | **61.20%** | 55.20% | **53.40%** | **67.50%** |
| | Pick-a-Pic V2 | Diff-DPO | SDXL | 68.00% | 77.80% | **71.20%** | 48.80% | **63.30%** | 61.40% |
| | | MaPO | SDXL | 66.50% | 72.40% | 53.70% | **67.50%** | 55.80% | 51.60% |
| | | Semi-DPO | SDXL | **74.00%** | **82.00%** | 65.00% | 65.30% | 57.20% | **70.00%** |
| | | Semi-DPO | Diff-DPO | **56.20%** | **55.20%** | 42.90% | **68.70%** | 44.00% | **60.50%** |
| | | Semi-DPO | MaPO | **63.90%** | **71.20%** | **62.90%** | 50.90% | **53.80%** | **69.30%** |

## 6.11 ADDITIONAL QUALITATIVE RESULTS

Figure 6: Qualitative comparison of Semi-DPO against baseline models

