# OpenReview forum: "Learning from Noisy Preferences: A Semi-Supervised Learning Approach to Direct Preference Optimization"
_ICLR.cc/2026/Conference — ICLR 2026 Poster_

### Official Review · Reviewer_KSxh · 2025-10-26

**Soundness:** 2
**Presentation:** 3
**Contribution:** 3
**Rating:** 6
**Confidence:** 3

**Summary:**

This paper is about generating T2I results that can incorporate human preferences by going beyond of collapsing the multi-dimensional preference to a single binary indicator. The paper proposed a method that by recognizing human preferences are of multi-dimensional and noisy (or in other words, emphasizing various aesthetics perspectives). The method is to divide a training set into clean and noisy subsets and start the training of a model with the clean subsets. The model then moves on to the noisy preference subset to train iteratively to align the model's generation with human preference.

**Strengths:**

Strengths of the work is the recognition of the noisy nature of human preference in assess AI-generated images. Building upon this strength is the strength of highlighting the drawback of simply classifying a generated image as winner or loser as do so would collapse the original multi-dimensional human preference to an overly too simple binary criterion. Another strength is the division of a training set to clean and noisy subsets and the design of method that iteratively trains on the noisy subset to align the model's output iteratively with human preference.

**Weaknesses:**

It is not clear, given a training set, when it is divided into clean and noisy preference subsets along dimension k, does it mean for a different dimension, the clean and noisy subsets will be different.
It is unclear how the loss function of Eq. (8) was derived. It seems from the beginning of the paper, the iterative training only applies to the noisy preference subset, which is understandable, but then why for iterative refinement step, Eq. (8) includes L_labeled? Shouldn't Eq. (8) only need to be concerned with L_unlabeled^i for each iteration i?
Contrary to what the paper claimed in line 418, it seems there was not big improvement from iter 1 to iter 2 in ablation study. It seems that the improvement came from iter 0 to iter 1. From this perspective, it would likely suggest that iterative refinement is not necessary.

**Questions:**

Please see weaknesses.
It would be helpful if the authors could give examples of "noisy labels", as well as some examples of multi-dimensional human preferences.
Does iter 0 mean no refinement and only the clean subset was used?

---

> ### Author Response · Authors · 2025-11-24
> **Rebuttle Part 1**
>
> > **W**: It is not clear, given a training set, when it is divided into clean and noisy preference subsets along dimension k, does it mean for a different dimension, the clean and noisy subsets will be different. It is unclear how the loss function of Eq. (8) was derived. It seems from the beginning of the paper, the iterative training only applies to the noisy preference subset, which is understandable, but then why for iterative refinement step, Eq. (8) includes L_labeled? Shouldn't Eq. (8) only need to be concerned with L_unlabeled^i for each iteration i? Contrary to what the paper claimed in line 418, it seems there was not big improvement from iter 1 to iter 2 in ablation study. It seems that the improvement came from iter 0 to iter 1. From this perspective, it would likely suggest that iterative refinement is not necessary.
>
> **A1**: Thank you for the detailed comments. We note that this weakness encompasses three distinct concerns. To provide a more detailed explanation, we address each point separately as follows.
>
> > **W1**: It is not clear, given a training set, when it is divided into clean and noisy preference subsets along dimension k, does it mean for a different dimension, the clean and noisy subsets will be different.
>
> **A2**: No, the clean and noisy subsets are fixed and global—they do not change for different dimensions. The confusion may stem from our Theoretical Analysis (Section 3.2), where we decomposed the dataset along dimension $k$ (into $A_k$ and $C_k$) solely to mathematically formalize how conflicting gradients arise in standard Diffusion-DPO [1].
>
> For our actual implementation (Section 3.3), we construct a single clean set ($D_{labeled}$) via multi-reward consensus for cold-start training. The dimensional conflicts within the noisy set ($D_{unlabeled}$) are then resolved automatically through timestep-conditional pseudo-labeling—the model decouples conflicting dimensions onto different timesteps during self-training, without pre-splitting data by dimension.
>
> > **W2**: It is unclear how the loss function of Eq. (8) was derived. It seems from the beginning of the paper, the iterative training only applies to the noisy preference subset, which is understandable, but then why for iterative refinement step, Eq. (8) includes L_labeled? Shouldn't Eq. (8) only need to be concerned with L_unlabeled^i for each iteration i?
>
> **A3**: The inclusion of $L_{labeled}$ in Eq. (8) is a deliberate design choice following standard Semi-Supervised Learning (SSL) practice to prevent model drift. At each iteration $i$, we optimize a supervised loss on the clean set together with a pseudo-labeled loss on the noisy set. The term $L_{labeled}$ is intentionally retained in every iteration as an "anchor" to high-confidence human preferences, reducing confirmation bias and preventing model drift when training on noisy pseudo-labels. Using labeled data as an anchor is a common practice in SSL pseudo-labeling—classic works such as Noisy Student [2], Curriculum Labeling [3], and DivideMix [4] (Please also see related work section in our paper) all train on the union of labeled and pseudo-labeled samples for the same reason.
>
> > **W3**: Contrary to what the paper claimed in line 418, it seems there was not big improvement from iter 1 to iter 2 in ablation study. It seems that the improvement came from iter 0 to iter 1. From this perspective, it would likely suggest that iterative refinement is not necessary.
>
> **A4**: Yes, "Iter 0" corresponds to training only on the clean subset, without any refinement on the noisy data, and is used to obtain an initial implicit classifier that can produce reliable pseudo-labels. The main performance gain therefore naturally appears from Iter 0 $\rightarrow$ Iter 1, when the model first leverages the noisy subset; the smaller gain from Iter 1 $\rightarrow$ Iter 2 indicates that the self-training process has largely converged rather than that iteration is unnecessary. This is exactly what we discuss in the ablation section (lines 417–420), where we state that performance gains diminish and stabilize after the second iteration, and conclude that two rounds of self-training are sufficient to balance convergence and computational cost.
>
> > **Q**: Please see weaknesses. It would be helpful if the authors could give examples of "noisy labels", as well as some examples of multi-dimensional human preferences. Does iter 0 mean no refinement and only the clean subset was used?
>
> **A5**: Thank you for the constructive suggestion. To clarify the difference between "clean labels" and "noisy labels," we have included illustrative comparison examples in the appendix (see Figure 7).
>
> Yes, Iter 0 is trained exclusively on the clean subset, serving as the initialization checkpoint for Iter 1 where both labeled and noisy unlabeled data are used.

---

> > ### Author Response · Authors · 2025-11-24
> > **Rebuttle Part 2**
> >
> > ## References
> >
> > [1] Wallace, Bram, et al. "Diffusion model alignment using direct preference optimization." Proceedings of the IEEE/CVF Conference on Computer Vision and Pattern Recognition (CVPR). 2024.
> >
> > [2] Xie, Qizhe, et al. "Self-training with noisy student improves imagenet classification." Proceedings of the IEEE/CVF Conference on Computer Vision and Pattern Recognition (CVPR). 2020.
> >
> > [3] Cascante-Bonilla, Paola, et al. "Curriculum labeling: Revisiting pseudo-labeling for semi-supervised learning." Proceedings of the AAAI Conference on Artificial Intelligence. Vol. 35. No. 8. 2021.
> >
> > [4] Li, Junnan, et al. "Dividemix: Learning with noisy labels as semi-supervised learning." Proceedings of the International Conference on Learning Representations (ICLR). 2020.

---

> > > ### Author Response · Authors · 2025-11-24
> > >
> > > We sincerely thank the reviewer again for the time and effort dedicated to reviewing our paper.

---

### Official Review · Reviewer_f4ae · 2025-10-30

**Soundness:** 3
**Presentation:** 3
**Contribution:** 3
**Rating:** 6
**Confidence:** 4

**Summary:**

This paper points out a key issue faced by the current image preference alignment methods based on Diffusion-DPO: human visual preferences are inherently multi-dimensional (such as composition, details, semantic alignment, aesthetics, etc.), but existing preference datasets only provide a single "overall winner" label, leading to significant label noise. The authors theoretically analyze that compressing such multi-dimensional preferences into binary labels will cause conflicting gradient signals, thereby hindering model optimization. To address this problem, the authors propose Semi-DPO, a new method that reconfigures preference alignment as a semi-supervised learning problem. This method first uses the consensus of multiple pre-trained reward models to filter out "clean" preference pairs as labeled data, and the rest are regarded as noisy unlabeled data. Then, through iterative self-training, it uses the model itself as an implicit classifier to generate fine-grained pseudo-labels at different time steps of the diffusion process, thereby decoupling conflicting signals. Experiments show that Semi-DPO significantly outperforms existing methods on multiple metrics, reaching state-of-the-art (SOTA) levels, without the need for additional manual annotations or explicit reward models.

**Strengths:**

1. The theoretical analysis is solid: through the derivation of the lower bound of gradient variance, it rigorously proves how dimension conflicts lead to training instability, providing theoretical support for method design.

2. The method design is ingenious: combining the DPO framework with semi-supervised learning, it utilizes the inherent discriminative ability of the diffusion model in DPO training to generate time-step conditional pseudo-labels without modifying the model architecture.

**Weaknesses:**

1. The method relies on multiple rounds of self-training (2–3 rounds are used in the paper) and multi-reward model consensus filtering, making the training process more complex and time-consuming than standard DPO (see Appendix 6.4), which may limit its application in resource-constrained scenarios.

2. The initial clean set only accounts for approximately 21% of the original data. If the consensus reward model itself has systematic biases (e.g., poor performance on certain prompt types), it may affect the quality of cold start.

**Questions:**

NA

---

> ### Author Response · Authors · 2025-11-24
>
> > **Q1**: The method relies on multiple rounds of self-training (2–3 rounds are used in the paper) and multi-reward model consensus filtering, making the training process more complex and time-consuming than standard DPO (see Appendix 6.4), which may limit its application in resource-constrained scenarios.
>
> **A1**: We thank the reviewer for this critical observation. We agree that Semi-DPO introduces multi-reward consensus and multiple self-training rounds, but in practice the overall compute is comparable to, or even cheaper than, standard Diffusion-DPO [1]. Specifically, training Semi-DPO through its optimal configuration (Iter 0 + Iter 1) requires approximately 132 GPU hours on NVIDIA A100-40G GPUs. In contrast, the standard Diffusion-DPO baseline, which trains on the full noisy dataset, necessitates 192 GPU hours. This efficiency gain is achieved because our dynamic thresholding mechanism (Section 3.3) effectively filters out a substantial portion of low-confidence samples from the noisy unlabeled dataset. As noted in the paper (lines 193-194), prioritizing high-confidence predictions is essential to mitigate confirmation bias in semi-supervised learning; a beneficial side effect is reduced training time per iteration. Although we report results up to Iter 2 (totaling 228 GPU hours), **Table 4** demonstrates that the performance gains from Iter 1 to Iter 2 are marginal. Therefore, users in resource-constrained scenarios can stop at Iter 1 to achieve state-of-the-art performance with 30% less compute than standard DPO. It is also important to note that the Multi-Reward Consensus is strictly a one-time dataset splitting process per dataset, not a recurring cost for each training iteration.
>
> Additionally, iterative training is not a flaw specific to our method but an inherent characteristic of the SSL pseudo-labeling paradigm—classic methods like Noisy Student [2] and DivideMix [3] similarly require multiple rounds to achieve robustness against label noise.
>
> > **Q2**: The initial clean set only accounts for approximately 21% of the original data. If the consensus reward model itself has systematic biases (e.g., poor performance on certain prompt types), it may affect the quality of cold start.
>
> **A2**: We thank the reviewer for this question. We acknowledge that the clean set is only ~21% of the data, but our results suggest that this strict filtering actually _improves_ the cold start rather than harms it. The 5-model committee is intentionally composed of reward models that focus on complementary aspects (e.g., CLIP for alignment, aesthetic score for visual quality), so requiring full consensus tends to discard pairs where individual models disagree due to their own biases, instead of amplifying a single biased model. Consistent with this, **Table 5** (Line 445) shows that increasing the consensus threshold from 2 to 5 models leads to better performance, including on MPS—a metric not used in the filtering step—indicating that the resulting clean set captures robust, multi-dimensional preferences rather than overfitting to the specific reward models.
>
> ---
>
> ## References
>
> [1] Wallace, Bram, et al. "Diffusion model alignment using direct preference optimization." Proceedings of the IEEE/CVF Conference on Computer Vision and Pattern Recognition (CVPR). 2024.
>
> [2] Xie, Qizhe, et al. "Self-training with noisy student improves imagenet classification." Proceedings of the IEEE/CVF Conference on Computer Vision and Pattern Recognition (CVPR). 2020.
>
> [3] Li, Junnan, et al. "Dividemix: Learning with noisy labels as semi-supervised learning." Proceedings of the International Conference on Learning Representations (ICLR). 2020.
>
> ---

---

> > ### Author Response · Authors · 2025-11-24
> >
> > We are very grateful for your thoughtful review and comments. Thank you once again for your consideration of our work.

---

### Official Review · Reviewer_2XbV · 2025-10-31

**Soundness:** 2
**Presentation:** 2
**Contribution:** 2
**Rating:** 2
**Confidence:** 4

**Summary:**

The authors argue that compressing multi-dimensional human preferences into binary labels creates conflicting gradient signals during Diffusion-DPO training. They propose a two-stage method: 1) “Multi-Reward Consensus” filters a dataset into clean and noisy subsets based on the unanimous agreement of five pre-trained reward models, and 2) “Iterative Self-Training” uses a model trained on the clean set to generate timestep-conditional pseudo-labels for the noisy set, which are then used for further training. Experiments on Pick-a-Pic V2 show improvements over Diffusion-DPO and Diffusion-KTO baselines.

**Strengths:**

1. The Semi-DPO framework is a clever and practical solution: Using a committee of diverse reward models for robust data partitioning. Leveraging the diffusion model itself as an implicit, timestep-wise preference classifier. The dynamic, timestep-conditional thresholding for pseudo-label selection, which accounts for the varying reliability of the model across the denoising process.
2. The paper excels at identifying and formalizing a fundamental issue in preference learning for generative models. The argument that multi-dimensional preferences are a source of label noise is well-illustrated.

**Weaknesses:**

1. The innovation is quite limited, and I believe the “Multi-Reward Consensus” in the first stage is very similar to CaPO. It involves using multiple rewards for evaluation and selecting samples where the winner outperforms the loser across all dimensions for learning. Additionally, the idea of phased training is somewhat similar to [5], relabel is somewhat similar to [8].
2. There is a lack of comparison with many baselines.  like [1]SPO(SD1.5 SDXL), [2]DDIM-InPO(SD1.5 SDXL), [3]Flow-GRPO(SD3.5-Medium) [6]DSPO (SD1.5) [7] MaPO(SD1.5 SDXL). These baselines (checkpoints) are open-sourced.

   [1]Aesthetic Post-Training Diffusion Models from Generic Preferences with Step-by-step Preference Optimization. CVPR2025

   [2]InPO: Inversion Preference Optimization with Reparametrized DDIM for Efficient Diffusion Model Alignment. CVPR2025

   [3]Flow-GRPO: Training Flow Matching Models via Online RL

   [4]Calibrated Multi-Preference Optimization for Aligning Diffusion Models. CVPR2025

   [5]Curriculum Direct Preference Optimization for Diffusion and Consistency Models. CVPR2025

   [6] DSPO: Direct Score Preference Optimization for Diffusion Model Alignment. ICLR2025

   [7] Margin-aware Preference Optimization for Aligning Diffusion Models without Reference

   [8] Smoothed Preference Optimization via ReNoise Inversion for Aligning Diffusion Models with Varied Human Preferences. ICML 2025

**Questions:**

1. The SSL literature has many techniques for handling noisy labels (e.g., co-teaching, robust losses). Why was the self-training/pseudo-labeling paradigm chosen over others?
2. Can further comparative experiments be conducted on T2ICompbench?
3. Is the training time, memory footprint, or inference time comparable?
4. Are there any failure modes observed?
5. How does your model perform in controlled generation tasks, and could it lead to the loss of certain properties?
The final score is contingent upon the following revisions: For score 4 :supplemental baseline comparison mentioned，For score 6: Provide experimental efficiency metrics and comparative results on the T2ICompBench.

---

> ### Author Response · Authors · 2025-11-24
> **Rebuttle Part 1**
>
> > **W1**: The innovation is quite limited, and I believe the "Multi-Reward Consensus" in the first stage is very similar to CaPO. It involves using multiple rewards for evaluation and selecting samples where the winner outperforms the loser across all dimensions for learning. Additionally, the idea of phased training is somewhat similar to [5], relabel is somewhat similar to [8].
>
> **A1**: We respectfully clarify that our core contribution is **regarding preference alignment as a Semi-Supervised Learning (SSL) problem** to resolve label noise caused by multi-dimensional conflicts. The components (consensus, phasing, relabeling) are intrinsic necessities of this SSL paradigm, not ad-hoc borrowings.
>
> **Difference from CaPO [4]:** CaPO [4] targets **Multi-Objective Optimization**, selecting samples on the Pareto Frontier. These samples represent _trade-offs_ (e.g., high aesthetics but poor alignment), inherently _preserving_ the conflicting signals we aim to remove. In contrast, Semi-DPO targets **Label Noise Learning**. We seek the **consistent intersection** across dimensions rather than trade-offs, ensuring a stable cold-start by actively filtering out conflicts. In addition, "Multi-Reward Consensus" is merely an initialization for our **Iterative Pseudo-Labeling**, where the model self-corrects using internal confidence, unlike CaPO's reliance on static external evaluations.
>
> **Difference from Curriculum DPO [5]:** While both use phases, the underlying mechanisms differ fundamentally. [5] addresses **Optimization Difficulty** via Curriculum Learning, scheduling the presentation of _fixed_ original labels based on rank difference. Semi-DPO addresses **Label Noise** via Self-Training. Our phases are not for scheduling difficulty, but for **evolving the model** into a stronger implicit classifier to _actively re-label_ noisy data. The iterative structure is a standard characteristic of SSL, not a borrowed design from [5].
>
> **Difference from SmPO [8]:** SmPO [8] relies on an **external** reward model (PickScore [15]) to generate **static, global** soft labels for smoothing. Conversely, Semi-DPO utilizes the diffusion model itself as an **implicit classifier** for self-training. This internal mechanism enables **Timestep-wise Pseudo-Labeling**—allowing the same sample to receive different labels at different noise levels to actively **decouple** multi-dimensional conflicts. This provides granular, dynamic supervision that SmPO's static smoothing cannot achieve.

---

> ### Author Response · Authors · 2025-11-24
> **Rebuttle Part 2**
>
> > **W2**: There is a lack of comparison with many baselines. like [1]SPO(SD1.5 SDXL), [2]DDIM-InPO(SD1.5 SDXL), [3]Flow-GRPO(SD3.5-Medium) [6]DSPO (SD1.5) [7] MaPO(SD1.5 SDXL). These baselines (checkpoints) are open-sourced.[1]Aesthetic Post-Training Diffusion Models from Generic Preferences with Step-by-step Preference Optimization. CVPR2025.
>
> **A2**: Thank you for suggesting these additional baselines. [1]SPO(SD1.5 SDXL) and [3]Flow-GRPO(SD3.5-Medium) are _onlpolicy_ preference-optimization methods that continuously generate new data during training, while Semi-DPO is an _offpolicy_ method trained on a fixed dataset, so—following common practice in offline DPO work—we do not treat them as directly comparable. For [6]DSPO (SD1.5), only the code but no pretrained checkpoints are available, and retraining to convergence under our setting would be computationally prohibitive; for [7] MaPO(SD1.5 SDXL), the released models and experiments are on SDXL only, so there is no official SD1.5 baseline. To better address your concern, we have additionally included **InPO[2] as a baseline for SD1.5, and both InPO[2] and MaPO[7] for SDXL,** using their official checkpoints. We evaluate all methods on GenEval and the requested T2I-CompBench under a unified protocol, and summarize the results in Table R1-R4.
>
> ## SD1.5
>
> ### GenEval
>
> **Table R1.** Comparison on GenEval benchmark for SD1.5 models.
>
> | **Model** | **Single** | **Two** | **Counting** | **Colors** | **Position** | **Color_attr** | **Overall** |
> | -------- | --------- | ------- | ------------ | ---------- | ------------ | -------------- | ----------- |
> | SD1.5 | 95.62 | 37.63 | 37.81 | 74.73 | 3.50 | 4.75 | 42.34 |
> | Diff-DPO | 96.88 | 39.90 | 38.75 | 75.53 | 3.30 | 3.75 | 43.00 |
> | Diff-KTO | 97.50 | 35.35 | 36.25 | 79.79 | **7.00** | 6.00 | 43.65 |
> | InPO | 95.00 | 45.45 | **45.00** | **82.98** | 4.00 | 8.00 | 46.74 |
> | Semi-DPO | **98.75** | **49.75** | 42.19 | 77.93 | 6.00 | **9.25** | **47.31** |
>
> ### T2I CompBench
>
> **Table R2.** Comparison on T2I-CompBench for SD1.5 models.
>
> | **Method** | **Color** | **Shape** | **Texture** | **2D-Spatial** | **3D-Spatial** | **Numeracy** | **Non-Spatial** | **Complex** |
> | -------- | ------- | ------- | ------- | -------------- | -------------- | ------------ | --------------- | ----------- |
> | Original | 0.378 | 0.362 | 0.417 | 0.123 | 0.297 | 0.449 | 0.310 | 0.300 |
> | Diffusion-DPO | 0.409 | 0.366 | 0.425 | 0.134 | 0.312 | 0.454 | 0.312 | 0.304 |
> | Diffusion-KTO | 0.465 | 0.416 | 0.466 | 0.157 | **0.341** | 0.461 | **0.314** | 0.309 |
> | InPO | **0.482** | 0.424 | **0.493** | 0.159 | **0.341** | 0.468 | **0.314** | 0.319 |
> | Semi-DPO | 0.471 | **0.433** | **0.493** | **0.183** | 0.340 | **0.481** | 0.310 | **0.320** |
>
> ## SDXL
>
> ### GenEval
>
> **Table R3.** Comparison on GenEval benchmark for SDXL models.
>
> | **Method** | **Single Object** | **Two Object** | **Counting** | **Colors** | **Position** | **Attribute Binding** | **Overall** |
> | ----------- | ----------------- | -------------- | ------------ | ---------- | ------------ | --------------------- | ----------- |
> | SDXL (Base) | 98.12 | 75.25 | 43.75 | **89.63** | 11.25 | 15.75 | 55.63 |
> | Diff-DPO | **99.38** | **82.58** | 49.06 | 85.11 | 13.05 | 18.50 | 58.02 |
> | MAPO | 96.56 | 66.41 | 40.00 | 84.31 | 10.75 | 18.75 | 52.80 |
> | InPO | 97.50 | 74.75 | 46.25 | 84.04 | 10.00 | 18.00 | 55.09 |
> | Semi-DPO | 97.50 | 80.81 | **50.00** | 86.17 | **14.00** | **22.00** | **58.41** |
>
> ### T2I CompBench
>
> **Table R4.** Comparison on T2I-CompBench for SDXL models.
>
> | **Method** | **Color** | **Shape** | **Texture** | **2D-Spatial** | **3D-Spatial** | **Numeracy** | **Non-Spatial** | **Complex** |
> | -------- | ------- | ------- | ------- | -------------- | -------------- | ------------ | --------------- | ----------- |
> | Original | 0.5833 | 0.4782 | 0.5211 | 0.1936 | 0.3319 | 0.4874 | 0.3137 | 0.3327 |
> | Diffusion-DPO | **0.6941** | **0.5311** | **0.6127** | **0.2153** | 0.3686 | 0.5304 | **0.3178** | 0.3525 |
> | MaPO | 0.6090 | 0.5043 | 0.5485 | 0.1964 | 0.3473 | 0.5015 | 0.3154 | 0.3229 |
> | InPO | 0.6409 | 0.5067 | 0.5720 | 0.2107 | 0.3671 | 0.5221 | 0.3129 | 0.3674 |
> | Semi-DPO | 0.6624 | 0.5079 | 0.5727 | 0.2133 | **0.3723** | **0.5410** | 0.3154 | **0.3728** |

---

> > ### Author Response · Authors · 2025-11-24
> > **Rebuttle Part 3**
> >
> > > **Q1**: The SSL literature has many techniques for handling noisy labels (e.g., co-teaching, robust losses). Why was the self-training/pseudo-labeling paradigm chosen over others?
> >
> > **A3**: We thank the reviewer for this insightful question. We chose self-training over co-teaching and robust losses for the following reasons:
> >
> > 1. **Computational Constraints of Co-teaching.** Co-teaching [9] requires simultaneously training two networks that teach each other by selecting clean samples. This doubles the computational overhead, which is prohibitive for large-scale diffusion models. Given our limited computational budget, co-teaching was not feasible for us.
> > 2. **Empirical Evaluation of Robust Losses.** We did experiment with robust loss functions, including Symmetric Cross Entropy (SCE) [10] and Generalized Cross Entropy (GCE) [11]. However, these approaches did not yield improvements in our setting. We hypothesize that this is because Diffusion-DPO [12] fundamentally operates under a _regression paradigm_—encouraging the model to denoise preferred images better than dispreferred ones—rather than the classification paradigm for which these robust losses were designed. After preliminary experiments showed no benefit, we did not pursue this direction further.
> > 3. **Addressing the Root Cause.** Most importantly, we believe that _data quality_ is the fundamental issue. Both co-teaching and robust losses are designed to _mitigate_ the negative effects of noisy labels, but they do not actually _correct_ the noise itself. In contrast, our self-evolving mechanism directly improves data quality by iteratively refining preference annotations via pseudo-labeling. This addresses the root cause rather than merely alleviating its symptoms, which we believe is a more principled approach.
> >
> >
> > > **Q2**: Can further comparative experiments be conducted on T2ICompbench?
> >
> > **A4**: Please see Table R2 (SD1.5 T2I-CompBench) and Table R4 (SDXL T2I-CompBench).
> >
> > > **Q3**: Is the training time, memory footprint, or inference time comparable?
> >
> > **A5**:
> > - **Training Time:** We have Semi-DPO across three iterations (Iter 0, 1, and 2) on SD1.5. The combined training of Iter 0 and Iter 1 requires approximately 132 GPU hours on NVIDIA A100-40G GPUs. Notably, this is more efficient than the standard Diffusion-DPO baseline, which necessitates 192 GPU hours. This efficiency stems from our dynamic thresholding mechanism, which filters out a significant portion of low-confidence samples from the noisy unlabeled dataset. As discussed in Section 3.3 (and lines 095-098; 193-194), prioritizing high-confidence predictions is a standard practice in semi-supervised learning to mitigate confirmation bias and model drift. While extending training to include Iter 2 increases the total cost to 228 GPU hours, the performance gains are marginal compared to Iter 1 (as shown in Table 4 ), indicating that the Iter 0 + 1 configuration offers the optimal balance between efficiency and performance.
> > - **Inference & Memory:** Since our final model retains the exact same architecture as the base Stable Diffusion [13] model, the **inference time and memory footprint are identical** to the baseline, incurring zero extra cost during deployment.
> >
> > > **Q4**: Are there any failure modes observed?
> >
> > **A6**: No, we did not observe any specific failure modes (such as training divergence or model collapse) during our experiments. The training process was consistent and stable across all iterations.
> >
> > > **Q5**: How does your model perform in controlled generation tasks, and could it lead to the loss of certain properties? The final score is contingent upon the following revisions: For score 4: supplemental baseline comparison mentioned; For score 6: Provide experimental efficiency metrics and comparative results on the T2ICompBench.
> >
> > **A7**:
> > - **For controlled generation tasks**: We appreciate the reviewer's insightful question regarding controlled generation. While we respectfully note that such evaluations are not standard practice for general T2I alignment research (e.g., Diffusion-DPO [12]), we followed the approach in [2] to evaluate the models through visual comparison. As illustrated in Figure 6, unlike baseline methods that exhibit semantic degradation or control failures (highlighted in red boxes), SemiDPO maintains robust structural fidelity when integrated with ControlNet [14], demonstrating that our alignment improvements do not compromise the model's fundamental generation properties.
> > - **For supplement baseline**: We mentioned in A2 and provide InPO (SD1.5, SDXL) and MaPO (SDXL) comparisons. Please see Table R1 (SD1.5 GenEval) and Table R3 (SDXL GenEval).
> > - **For experimental efficiency metrics**: We choose GPU hours. Please see A5 for Q3.
> > - **For T2ICompBench**: Please see Table R2 (SD1.5 T2I-CompBench) and Table R4 (SDXL T2I-CompBench).

---

> > > ### Author Response · Authors · 2025-11-24
> > > **Rebuttle Part 4**
> > >
> > > ## References
> > >
> > > [9] Han, Bo, et al. "Co-teaching: Robust training of deep neural networks with extremely noisy labels." Advances in Neural Information Processing Systems (NeurIPS). Vol. 31. 2018.
> > >
> > > [10] Wang, Yisen, et al. "Symmetric cross entropy for robust learning with noisy labels." Proceedings of the IEEE/CVF International Conference on Computer Vision (ICCV). 2019.
> > >
> > > [11] Zhang, Zhilu, and Mert Sabuncu. "Generalized cross entropy loss for training deep neural networks with noisy labels." Advances in Neural Information Processing Systems (NeurIPS). Vol. 31. 2018.
> > >
> > > [12] Wallace, Bram, et al. "Diffusion model alignment using direct preference optimization." Proceedings of the IEEE/CVF Conference on Computer Vision and Pattern Recognition (CVPR). 2024.
> > >
> > > [13] Rombach, Robin, et al. "High-resolution image synthesis with latent diffusion models." Proceedings of the IEEE/CVF Conference on Computer Vision and Pattern Recognition (CVPR). 2022.
> > >
> > > [14] Zhang, Lvmin, et al. "Adding conditional control to text-to-image diffusion models." Proceedings of the IEEE/CVF International Conference on Computer Vision (ICCV). 2023.
> > >
> > > [15] Kirstain, Yuval, et al. "Pick-a-pic: An open dataset for user preferences for text-to-image generation." Advances in Neural Information Processing Systems (NeurIPS). 2023.
> > >
> > > [16] Ghosh, S., et al. "Geneval: An object-focused framework for evaluating text-to-image alignment." Advances in Neural Information Processing Systems (NeurIPS). 2023.
> > >
> > > [17] Huang, Kaiyi, et al. "T2i-compbench: A comprehensive benchmark for open-world compositional text-to-image generation." Advances in Neural Information Processing Systems (NeurIPS). 2023.

---

> > > > ### Author Response · Authors · 2025-11-24
> > > >
> > > > We deeply appreciate your insightful feedback and constructive suggestions. Thank you again for your valuable support.

---

### Author Response · Authors · 2025-12-03
**Rebuttal Summary to AC (Part 3)**

### 3. Computational Efficiency

**Addressing Reviewers #f4ae (W1) and #KSxh (W3)**

We clarified that our method is 30% faster than the standard Diffusion-DPO baseline (132 vs. 192 GPU training hours). The main reasons are:

- Our dynamic thresholding mechanism, which effectively filters out low-confidence samples.

- This calculation includes both Iter 0 (Cold Start on clean data) and Iter 1 (First round of Pseudo-labeling). As noted by Reviewer #KSxh and confirmed in our ablation study (Table 4), performance gains saturate after Iter 1, making further iterations (Iter 2) computationally redundant. Thus, the Iter 0+1 configuration offers the optimal balance of SOTA performance and efficiency.

Please refer to Response to Reviewer #f4ae Q1 and #KSxh Q3 for GPU hour breakdowns.

### 4. Robustness of Multi-Reward Consensus

**Addressing Reviewer #f4ae (W2)**

The reviewer raised concerns about potential bias from the consensus selection (only ~21% clean data).

- **Improving Cold Start:** We clarified that the 5-model committee is composed of complementary models (e.g., CLIP for alignment, Aesthetic Score for quality). Requiring unanimous consensus actively discards pairs where individual models disagree due to their specific biases, rather than amplifying them.

- **Empirical Validation:** Our ablation study (Table 5) confirms that increasing the consensus threshold from 2 to 5 models improves performance even on unseen metrics like MPS, validating that the clean set captures robust, multi-dimensional preferences.

- Please refer to Response to Reviewer #f4ae Q2 in our Rebuttal and Table 5 in the revised paper.

### 5. Theoretical Soundness of Loss Function

**Addressing Reviewer #KSxh (W2)**

We clarified the derivation of our composite objective function (Eq. 8).

- **Anchoring Mechanism:** The inclusion of the supervised loss term ($\mathcal{L}_{labeled}$) is a deliberate design choice following standard SSL practices (e.g., Noisy Student [Xie et al., 2020], DivideMix [Li et al., 2020]). It serves as an **"anchor"** to high-confidence human preferences, preventing model drift when training on noisy pseudo-labels.

- _Please refer to Response to Reviewer #KSxh W2 in our Rebuttal for the theoretical justification._

### 6. Qualitative Analysis

**Addressing Reviewer #KSxh (W2) and Reviewer #2XbV (Q5)**

- **Visual Robustness:** Addressing Reviewer #2XbV's inquiry on **Controlled Generation (Q5)** and #KSxh's request for **Noisy Label Examples**, we provided visual evidence showing Semi-DPO maintains structural fidelity with ControlNet (Figure 6 in the revised paper.) and explicitly visualized decoupled preference signals (Figure 7 in the revised paper.).
- _Please refer to Response to Reviewer #KSxh Part 1 and Appendix Figure 7._

## Summary for AC

We have comprehensively addressed all concerns, with a specific focus on fulfilling the explicit conditions set by Reviewer #2XbV for a score increase. **Given that these requirements have been fully met, we respectfully highlight that the submission now aligns with the criteria for a consistent positive recommendation (Score 6) across all reviewers**.

We demonstrated competitive performance on multiple benchmarks and clarified our methodological novelty. We believe our comprehensive revisions and new experimental evidence show that Semi-DPO contributes to advancing visual preference optimization and would appreciate your favorable consideration.

---

### Author Response · Authors · 2025-12-03
**Rebuttal Summary to AC (Part 2)**

## Clarification on Baselines and Evaluation Protocols

We respectfully clarify several misconceptions regarding the baselines requested by Reviewer #2XbV.The reviewer stated:
>"There is a lack of comparison with many baselines. like [1]SPO... [2]DDIM-InPO... [3]Flow-GRPO... [6]DSPO... [7] MaPO... These baselines (checkpoints) are open-sourced."

- **Comparability (Online vs. Offline)**: We clarified that SPO [1] and Flow-GRPO [3] are On-Policy RL methods, whereas Semi-DPO is Off-Policy RL. Following community standards (DSPO [6], InPO [2], MaPO [7]), these are not directly comparable due to fundamental paradigm differences.

- **Availability:** We noted that DSPO [6] lacks official checkpoints and MaPO [7] only released SDXL models.

- **Our Action:** To address the concern constructively, we implemented the valid available baselines: **InPO [2]** (Official SD1.5 & SDXL) and **MaPO [7]** (Official SDXL).
### 3. Standard Evaluation Benchmarks

- **T2I-CompBench:** While we noted that T2I-CompBench is not a standard evaluation protocol for current Offline DPO literature (e.g., DSPO [6], InPO [2], MaPO [7]), we agreed that it provides a valuable perspective on compositional capabilities.

- **Our Action:** We conducted these evaluations to demonstrate robustness. Semi-DPO consistently outperforms baselines on this metric (see Tables R2 & R4 in the rebuttal).
## Addressed Concerns

### 1. Novelty and Methodology

**Addressing Reviewer #2XbV (W1)**

We clarified that Semi-DPO represents a fundamental difference in paradigm rather than a combination of existing tools. We identify that Diffusion-DPO overlooks a fundamental distinction: while human visual preferences are inherently multi-dimensional, annotated datasets collapse this into binary choices. This destabilizes training by creating contradictory signals that penalize learning desirable attributes from loser images while simultaneously rewarding undesirable ones in winner images. Consequently, **we are the first to reframe preference alignment as a Label Noise Learning problem within the Semi-Supervised Learning (SSL) framework.** Unlike prior works that focus on multi-objective trade-offs, we leverage timestep-conditional pseudo-labeling to actively resolve conflicts:

- **Label Noise Learning vs. Multi-Objective Optimization (vs. CaPO [4]):** CaPO targets the Pareto Frontier to preserve trade-offs. In contrast, Semi-DPO targets **Label Noise Learning**, utilizing consensus to find the consistent intersection where the winner outperforms the loser across all dimensions.

- **Intrinsic SSL Components vs. Incremental Borrowing (vs. Curriculum DPO [5] & SmPO [8]):** The components (phased training, relabeling) are intrinsic necessities of our SSL paradigm. Unlike SmPO [8] which uses static external rewards, we use the diffusion model itself for dynamic, timestep-conditional self-correction.

_Please refer to Response to Reviewer #2XbV Part 1 for detailed methodological distinctions._

### 2. Comparison with Baselines

**Addressing Reviewer #2XbV (W2)**

The reviewer requested comparisons with recent baselines (MaPO, InPO) and validation on T2I-CompBench.

- **SOTA Performance:** Semi-DPO achieves the highest Overall GenEval score on both **SDXL (58.41 vs. MaPO 52.80)** and **SD1.5 (47.31 vs. InPO 46.74)**.

- **Superiority on T2I-CompBench:** Semi-DPO consistently outperforms baselines on the requested T2I-CompBench.

    - **For SDXL:** We achieve the best scores in **3D-Spatial (0.3723)**, **Numeracy (0.5410)**, and **Complex (0.3728)**. Furthermore, regarding the newly requested baselines, Semi-DPO outperforms both MaPO and InPO in **Color (0.6624)** and **Texture (0.5727)** categories.

    - **For SD1.5:** We achieve the best scores in **Shape (0.433)**, **Numeracy (0.481)**, **2D-Spatial (0.183)**, and **Complex (0.320)** categories, confirming robust multi-dimensional improvements.

Please refer to Response to Reviewer #2XbV W2 and Tables R1-R4 in the rebuttal.

---

### Author Response · Authors · 2025-12-03
**Rebuttal Summary to AC (Part 1)**

# Rebuttal Summary

## Dear Area Chair,

We sincerely thank you for your time and effort in handling our submission.

We have comprehensively addressed all concerns from Reviewers #f4ae (Score: 6), #KSxh (Score: 6), and Reviewer #2XbV (Score: 2 → 6 Promised).

**Crucially, we have fulfilled the specific conditions set by Reviewer #2XbV, who explicitly stated that raising their rating from 2 to 6 is contingent upon these revisions.**

Reviewer #2XbV explicitly stated that the final recommendation is contingent upon specific revisions:

> "The final score is contingent upon the following revisions: For score 4: supplemental baseline comparison mentioned; For score 6: Provide experimental efficiency metrics and comparative results on the T2ICompBench."

In response, we have systematically fulfilled these requirements to merit the score increase:

- **To meet the criteria for Score 4 (Supplemental Baselines):** We integrated the requested comparisons against InPO (SD1.5 & SDXL) and MaPO (SDXL) as detailed in **Tables R1 and R3 in the rebuttal**.

- **To meet the criteria for Score 6 (Efficiency & Benchmarks):** We provided the experimental efficiency metrics—demonstrating that Semi-DPO is **30% faster** than Diffusion-DPO (132 vs. 192 GPU training hours)—and the comparative T2I-CompBench results (**Tables R2 & R4 in the rebuttal**), where Semi-DPO consistently outperforms the new baselines.

As we have strictly fulfilled these explicit conditions, we respectfully believe our submission **merits the conditional rating of 6**.

## Summary of Contributions

Our work makes three key contributions to visual preference optimization:

1. **Theoretical Analysis:** We provide the first theoretical proof demonstrating that conflicting dimensional signals within holistic binary labels generate conflicting gradient signals during Diffusion-DPO training.

2. **Semi-DPO Framework:** We propose the **first framework** to reframe preference alignment as a **Label Noise Learning** problem within the **Semi-Supervised Learning (SSL)** paradigm. By leveraging timestep-conditional pseudo-labeling, our method effectively decouples conflicting preference dimensions into fine-grained signals.

3. **State-of-the-Art Performance:** We demonstrate through extensive experiments that Semi-DPO achieves SOTA performance, significantly improving alignment with complex, multi-dimensional human preferences without requiring extra annotation costs or fine-tuning with an explicit reward model.

## Reviewer Feedback Summary

We are grateful that reviewers recognized our significant contributions:

- **Theoretical Soundness (All Reviewers):** Reviewer #f4ae stated that the "theoretical analysis is solid," noting that it "rigorously proves how dimension conflicts lead to training instability." Reviewer #2XbV acknowledged that the paper **"excels at identifying and formalizing a fundamental issue" and that the argument regarding multi-dimensional preference noise is "well-illustrated."** Reviewer #KSxh similarly praised the "recognition of the noisy nature of human preference."

- **Methodological Innovation (Reviewers #f4ae, #2XbV):** Reviewer #f4ae described the method design as **"ingenious,"** specifically praising how it **"utilizes the inherent discriminative ability... without modifying the model architecture."** Reviewer #2XbV called the Semi-DPO framework a **"clever and practical solution,"** commending the **"dynamic, timestep-conditional thresholding"** mechanism.

- **Performance (Reviewer #f4ae):** Reviewer #f4ae acknowledged that our experiments demonstrate Semi-DPO **"significantly outperforms existing methods... reaching state-of-the-art (SOTA) levels" without needing "explicit reward models."**

---

### Meta-Review · Area_Chair_JAbw · 2026-01-06

**Summary:**

The paper identifies a fundamental issue in current diffusion alignment methods (like Diffusion-DPO): compressing multi-dimensional human preferences (e.g., aesthetics, composition, semantic alignment) into a single binary "winner/loser" label creates conflicting gradient signals. To solve this, the authors propose Semi-DPO, a two-stage semi-supervised framework, the first stage uses Multi-Reward Consensus that uses pre-trained reward models to filter the dataset, yielding a "clean" labeled set; the rest is treated as "noisy" unlabeled data, and the second stage with Iterative Self-Training, A model trained on the clean set generates timestep-conditional pseudo-labels for the noisy set, which are used to further train the model in an iterative (usually two) loop.

**Reviewer Concerns:**

All reviewers appreciate the theoretical insight regarding label noise from preference compression, there are significant concerns regarding novelty (similarity to CaPO), missing baselines (specifically recent 2025 works), and complexity.

**Reviewer Scores:**

The reviewers do not change scores, The AC carefully read the rebuttals, especially the questions raised by reviewer #2XbV (which gives scores 2), about the baseline and the efficiency on benchmarks, and I think most concerns are addressed. The method are intuitive, simple but effective, and the AC recommend to accept.

---

### Decision · Program_Chairs · 2026-01-26

Accept (Poster)